# The lipocone superfamily, a unifying theme in metabolism of lipids, peptidoglycan and exopolysaccharides, inter-organismal conflicts and immunity

**A Maxwell Burroughs[†], Gianlucca G Nicastro[†], L Aravind***

Division of Intramural Research, National Library of Medicine, National Institutes of Health, Bethesda, United States

## eLife Assessment

This **fundamental** study presents a **compelling** and comprehensive analysis of the newly defined Lipocone superfamily, offering unprecedented insights into the evolutionary origins of Wnt proteins. The authors provide evidence that this superfamily evolved from membrane proteins. The work is exemplary in its use of sequence analysis and structural modeling and will be of broad interest to researchers studying protein evolution and enzymology.
[Editors' note: this paper was reviewed by Review Commons.]

**\*For correspondence:**
aravind@ncbi.nlm.nih.gov

[†]These authors contributed equally to this work

**Competing interest:** The authors declare that no competing interests exist.

**Abstract** Wnt proteins are critical signaling molecules in developmental processes across animals. Despite intense study, their evolutionary roots have remained enigmatic. Using sensitive sequence analysis and structure modeling, we establish that the Wnts are part of a vast assemblage of domains, the Lipocone superfamily, defined here for the first time. It includes previously studied enzymatic domains like the phosphatidylserine synthases (PTDSS1/2) and the TelC toxin domain from *Streptococcus intermedius*, the enigmatic VanZ proteins, the animal Serum Amyloid A (SAA), and a further host of uncharacterized proteins in a total of 30 families. Although the metazoan Wnts are catalytically inactive, we present evidence for a conserved active site across this superfamily, versions of which are consistently predicted to operate on head groups of either phospholipids or polyisoprenoid lipids, catalyzing transesterification and phosphate-containing head group cleavage reactions. We argue that this superfamily originated as membrane proteins, with one branch (including Wnt and SAA) evolving into diffusible versions. By comprehensively analyzing contextual information networks derived from comparative genomics, we establish that they act in varied functional contexts, including regulation of membrane lipid composition, extracellular polysaccharide biosynthesis, and biogenesis of bacterial outer-membrane components, like lipopolysaccharides. On multiple occasions, members of this superfamily, including the bacterial progenitors of Wnt and SAA, have been recruited as effectors in biological conflicts spanning inter-organismal interactions and anti-viral immunity in both prokaryotes and eukaryotes. These findings establish a unifying theme in lipid biochemistry, explain the origins of Wnt signaling, and provide new leads regarding immunity across the tree of life.

## Introduction

The canonical Wnt signaling network is central to developmental decisions across animals relating to axis patterning, cell fate, cell migration and proliferation, and systems morphogenesis at many levels

(*Richards and Degnan, 2009*; *Jessen et al., 2008*; *Morata and Lawrence, 1977*; *Rijsewijk et al., 1987*; *Sharma, 1973*; *Nusse and Varmus, 1982*; *Nusse and Varmus, 1992*). Other crucial pathways, dubbed non-canonical Wnt signaling pathways, include those that regulate planar cell polarity and intracellular calcium levels (*Komiya and Habas, 2008*; *Segalen and Bellaïche, 2009*; *Slusarski and Pelegri, 2007*). With these roles in development and homeostasis, dysfunction of Wnt signaling is causally associated with a range of diseases, including diverse cancer types and type II diabetes (*Logan and Nusse, 2004*; *Welters and Kulkarni, 2008*). Wnt signaling networks are centered on the secreted Wnt proteins acting as both paracrine and autocrine diffusible, extracellular messenger molecules (*Christian, 2000*). Wnt proteins are ligands for the N-terminal, cysteine-rich CBD/Fz domains of the Frizzled class of G-protein coupled receptors (GPCRs) (*Schulte and Bryja, 2007*; *Bhanot et al., 1996*). Modifications of the Wnt proteins via palmitoleoylation and glycosylation at internal sites are associated with their secretion (*Takada et al., 2006*). Palmitoleoylation of Wnt occurs at a conserved serine residue and is also required for recognition by the Frizzled receptors (*Kurayoshi et al., 2007*). Binding of the Frizzled receptor by Wnt recruits the Disheveled (Dsh) protein to its cytoplasmic face, in turn triggering a bevy of downstream responses, resulting in β-catenin stabilization in canonical pathways (*Gao and Chen, 2010*; *Klingensmith et al., 1994*). When β-catenin concentrations cross a threshold, it is translocated into the nucleus, where it acts as a transcriptional coactivator, usually with an HMG domain transcription factor, to stimulate multiple transcriptional programs (*Kramps et al., 2002*; *van Tienen et al., 2017*; *Archbold et al., 2012*).

Despite its initial discovery over 40 years ago, the evolutionary origins of the Wnt protein have, until recently, been mysterious (*Holstein, 2012*). In 2020, our group reported the discovery of the first prokaryotic versions of the Wnt domain (*Burroughs and Aravind, 2020*). Using comparative genomics, we showed that these bacterial Wnt domains present contexts characteristic of toxins or effectors in biological conflict systems (*Burroughs and Aravind, 2020*; *Aravind et al., 2022*). Prompted by these initial observations, we set out to comprehensively identify and computationally characterize the evolutionary relationships of these newly identified Wnt homologs in an effort to understand their evolutionary history and predict their functions.

Consequently, we were able to unify the Wnt family with several other domains into a large superfamily described for the first time herein. These include two biochemically characterized families that were hitherto not known as Wnt homologs: the phosphatidylserine synthase (PTDSS1/2, EC: 2.7.8.29) (*Kuge et al., 1997*; *Kuge et al., 1991*; *Stone and Vance, 1999*) and the toxin domain of TelC from *Streptococcus intermedius* (*Whitney et al., 2017*). However, the majority of the families we unify are either reported for the first time or are functionally poorly understood, including the animal Serum Amyloid A (SAA) (*Sack, 2018*) and the vancomycin resistance protein VanZ families (*Arthur et al., 1995*; *Arthur et al., 1999*). Our comparative genomics analyses, paired with existing experimental evidence, suggest that the superfamily is broadly comprised of enzymes operating on lipid head groups (e.g. transesterification reactions) in a diversity of biochemical contexts, notably including the regulation of membrane composition, extracellular biopolymer metabolism, and as effectors in biological conflicts. Thus, we identify a unifying theme across diverse aspects of lipid metabolism.

## Results
### Identification of the structural core of the Wnt domain

Although the structure of Wnt was described over a decade prior (*Bazan et al., 2012*; *Janda et al., 2012*; *Chu et al., 2013*), its origins have been a mystery as it is phylogenetically restricted to Metazoa. Much attention has been focused on the three extended β-hairpins and a poorly structured loop extruding out of the core, their stabilizing cysteine residues, and the absolutely conserved serine residue, the site of palmitoleoylation (*Janda et al., 2012*; *Zhong et al., 2021*; *Figure 1A*). Our discovery of the first prokaryotic Wnt domains helped define its ancestral α-helical core, revealing the cysteine-rich extensions as Metazoa-specific insertions. Comparison of the core of the metazoan Wnt with AlphaFold structural models of the prokaryotic versions (*Burroughs and Aravind, 2020*) revealed a shared globular domain composed of five α-helices (*Figure 1A*). As the prokaryotic homologs retained just the conserved core of the Wnt proteins, we named these the minimal Wnt (Min-Wnt) family. The core helices of the Min-Wnt family contained absolutely conserved sequence motifs

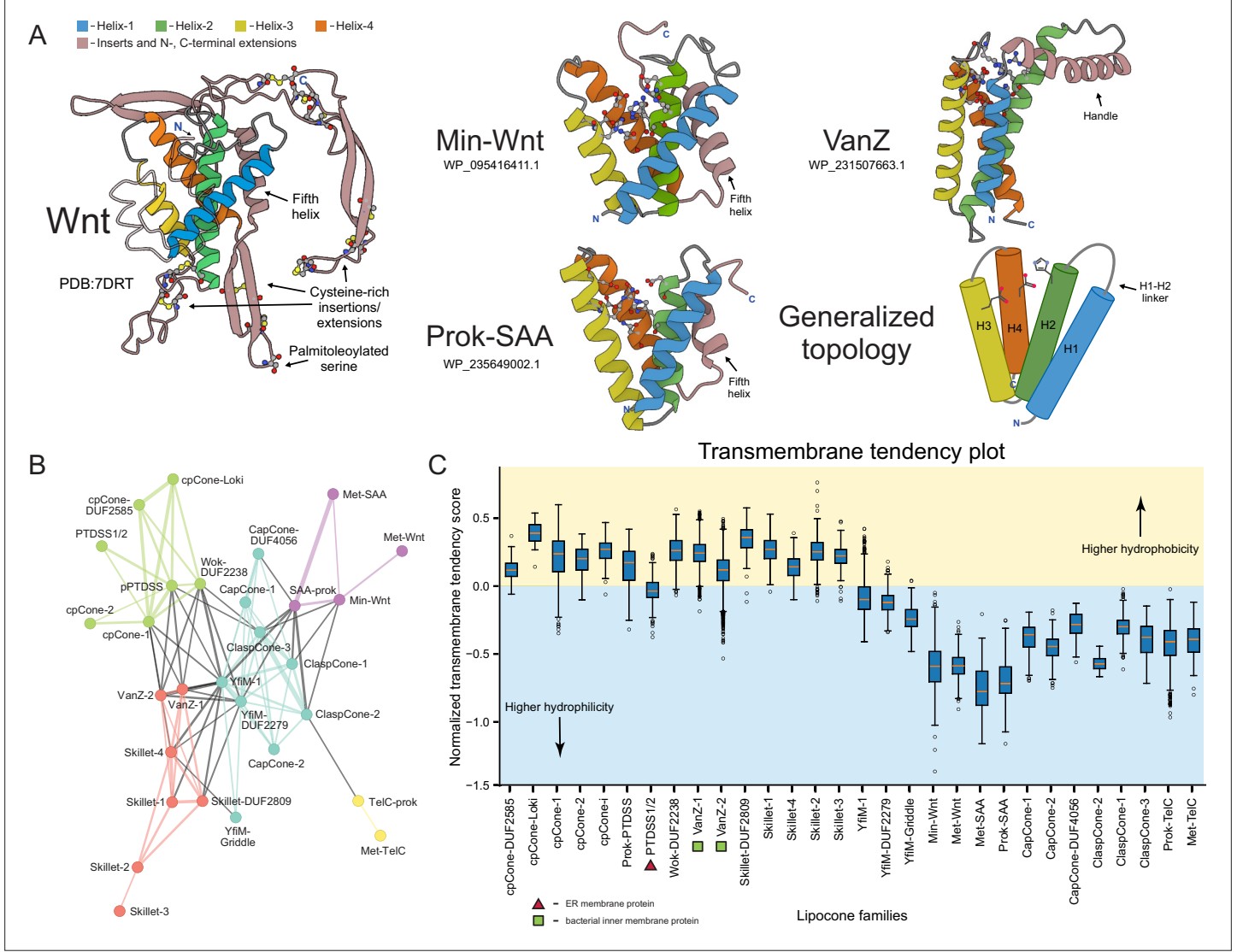

**Figure 1.** Identification and compositional analysis of the Lipocone superfamily. (**A**) The four individual helices forming the core of the Lipocone superfamily are consistently colored across the illustrated representatives. The inter-helix linkers are colored gray, and lineage-specific synapomorphic insertions and extensions are colored light brown. Active site and other residues of interest are rendered as ball-and-stick. Protein Data Bank (PDB) IDs or GenBank accessions used to generate AF3 models are provided. (**B**) Relationship network of the Lipocone families. The thickness of the edges is scaled by negative-log HHalign p-values. Families are colored according to the community identified by the Leiden algorithm (*Traag et al., 2019*) (see Methods). (**C**) Box plots displaying core helix transmembrane propensity scores of individual sequences within different Lipocone families. The horizontal divider represents the boundary between typical transmembrane (TM) and soluble sequences.

The online version of this article includes the following source data and figure supplement(s) for figure 1:

**Source data 1.** Transmembrane tendency scores by Lipocone family sequence for (**C**) and network YAML file for (**B**).

**Figure supplement 1.** Phyletic distribution patterns of Lipocone superfamily.

**Figure supplement 2.** Structural representatives of the Lipocone superfamily.

**Figure supplement 3.** Critical difference diagram depicting group-wise differences across transmembrane (TM) tendency score distributions in *Figure 1C* (see Methods).

**Figure supplement 4.** Structural diversity in the cpCone clade.

**Figure supplement 5.** Summary of the Methodology and Main Findings.

(*Figure 2*), consistent with the enzymatic function we had earlier proposed for them (*Burroughs and Aravind, 2020*) (see below).

Having defined this shared core, we initiated sequence-based homology searches in an effort to identify remote homologs. Iterative position-specific sequence matrix (PSSM)-based searches (see Methods) initially recovered animal and bacterial versions of the Serum Amyloid A (SAA) proteins, and further rounds of searching initiated from this set of sequences further recovered a vast collection of additional homologous families. As an example, a search initiated with a bacterial SAA-like sequence from *Bdellovibrio bacteriovorus* (Genbank acc: AHZ84906.1) retrieved a sequence overlapping with the Pfam models for 'Domain of unknown function,' DUF2279 (acc: WP_146898260.1, iteration: 5, e-value: 0.004), DUF4056 domain (acc: MBW8016507.1, iteration: 5, e-value: 0.005), and sequences automatically annotated as 'YfiM' in the GenBank database (acc: WP_019077413.1, iteration: 4, e-value: 0.004). Sequence profile-profile searches with HHpred confirmed these relationships and captured more distant ones. For instance, a HHpred search initiated with the *Bacteriovorax stolpii* Min-Wnt domain (acc: WP_102242990.1, residues 1–109) recovered the Pfam Wnt profile (PF: PF00110.23, p-value: 1.5e-6) and the Pfam SAA profile (PF: PF00277.22, p-value: 3.7e-5). Similarly, a HHpred search initiated with a Gemmatimonadetes sequence (acc: PYP94660.1, residues 75.170) recovered the DUF2279 Pfam profile (PF: PF10043.12, p-value: 5.4E-21), the DUF2238 profile (PF: PF09997.12, p-value: 1.3E-07), and the DUF4056 Pfam profile (PF: PF13265.9, p-value: 2.5E-05), among others.

Exhaustion of these searches, followed by clustering and manual inspection of the multiple sequence alignments of the retrieved sequences (see Methods), revealed a shared four-helix core across all of them, hereinafter referred to as H1 through H4 (*Figure 1A*). This 4-helix core of the domain was further confirmed by inspection of AlphaFold structural models constructed for representatives of the individual families, along with the rare instances of experimentally determined structures. These comparisons established that the above-mentioned fifth C-terminal helix in the Wnt core is a synapomorphy (shared derived character) restricted to the Wnts and closely related families like SAA (*Figure 1A*). In all, the results of our clustering analysis tallied 30 distinct families constituting a large superfamily. Remarkably, of these, 17 families had no pre-existing annotations. Phyletic analysis of individual families revealed a range of distributions, ranging from broad conservation in multiple superkingdoms of Life to those restricted to a small number of lineages (see below, *Figure 1—figure supplement 1*). A relationship network for the superfamily was constructed based on p-value and e-value scores using alignments of each family as a query in HHalign profile-profile searches against the rest (see Methods, *Figure 1B*). The Leiden community detection algorithm (*Traag et al., 2019*) was then applied to this network to identify higher-order assemblages (see Methods). These groupings were also supported by structural synapomorphies, such as a circular permutation and versions with a two-stranded 'handle' (see below).

The four helices conserved across the superfamily constitute a cone-like structure (*Figure 1A*), with the helices tending to coalesce on one end and opening out into a pocket on the other, lined by the conserved sequence positions (*Figure 1A*, *Figure 1—figure supplement 2*). The core is also marked by a linker between H1 and H2, which adopts characteristic extended conformations in certain families and higher-order groups. While the linkers joining H2 and H3 and H3 and H4 tend to be more constrained, there are some exceptions; for example, the extended loop insert housing the palmitoleoylated serine residue between H2 and H3 in the metazoan Wnt family (*Figure 1A*).

## Dramatic variability in hydrophobicity of the conserved core across the superfamily

We observed that these Wnt-related families dramatically varied in their hydrophobicity. Using an index for transmembrane propensity (*Zhao and London, 2006*) (see Methods) and comparing that to known transmembrane (TM) segments, we predict that the α-helices in 18 of the 30 families are hydrophobic enough to qualify as TM domains, and show a statistically significant tendency to group to the exclusion of the other families (*Figure 1C*, *Figure 1—figure supplement 3*). Thus, these are predicted to be integral membrane domains. Further, these 'hydrophobic families' often evince a broader and deeper phyletic distribution pattern than the less-hydrophobic families (*Figure 1—figure supplement 1*, methods), implying that the ancestral version of the superfamily was likely an integral membrane domain. Thus, their association with the lipid membrane, combined with the cone-like

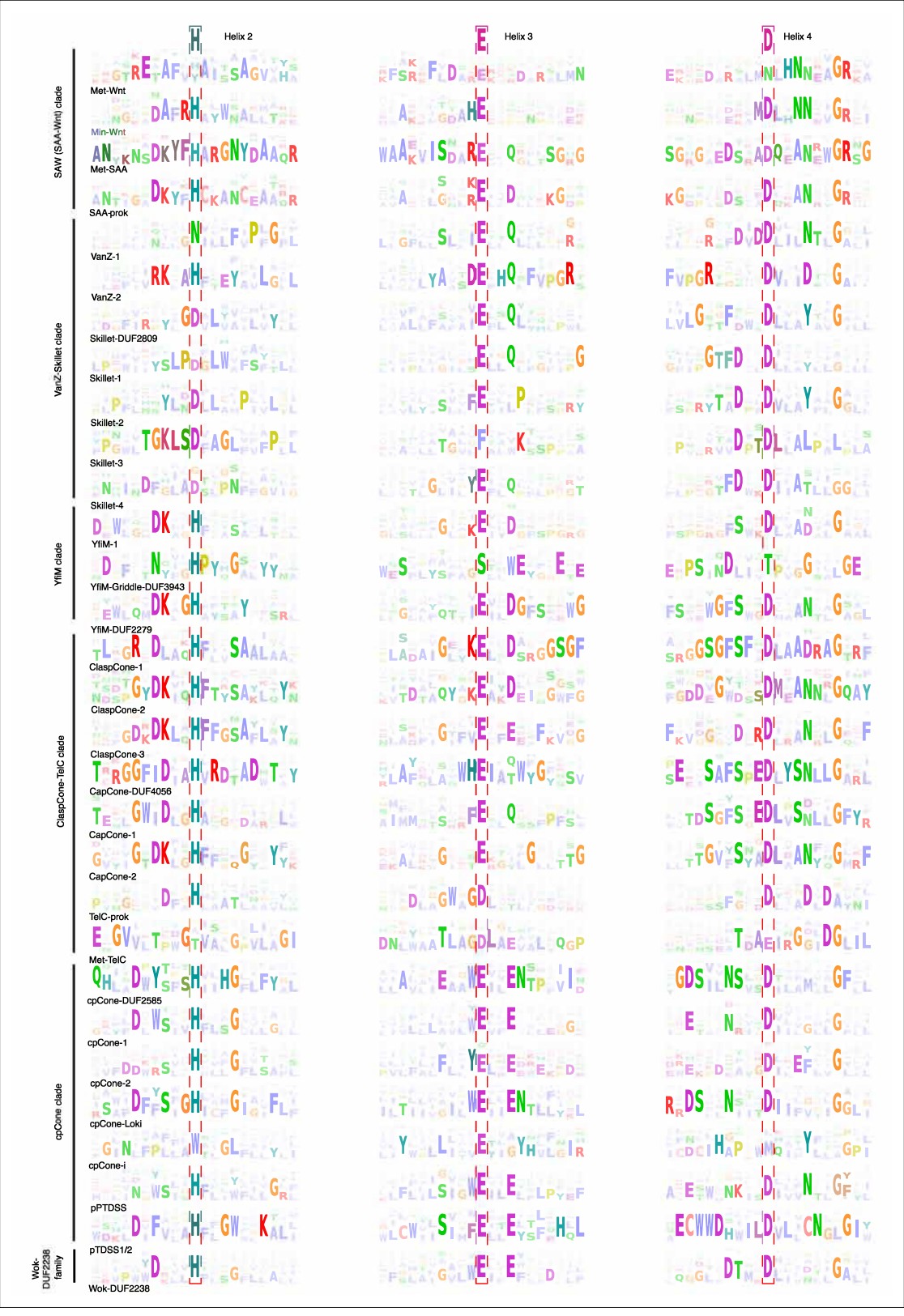

**Figure 2.** Sequence logo of conserved core elements of the Lipocone families. These correspond to the core helices H2, H3, and H4. The three conserved active site residue positions are boxed in dotted lines with the inferred ancestral residue indicated at the top of the alignment. Families are grouped and labeled on the left in their higher-order clades.

shape of the conserved core (*Figure 1A*), leads us to refer to the whole superfamily hereinafter as the *Lipocone* superfamily.

AlphaFold 3-assisted transmembrane topology prediction (*Jumper et al., 2021*) revealed that 14 of the 17 integral membrane families are consistently oriented with the aperture of the cone-like structure opening toward the outer face of the membrane. This predicted TM topology is also generally consistent with the domain fusions when present: e.g., domains that are typically cytoplasmic and those that have extracellular or periplasmic functions are, respectively, predicted as projecting either inside or outside the membrane (see below). However, three families in the cpCone clade (see below) did not yield consistent orientation predictions, potentially owing to the diversity of structural variations observed in the clade, including a circular permutation event.

## A unified biochemistry for the Lipocone superfamily

Of the 30 identified families, 26 display a striking conservation pattern of polar residues associated with the pocket of the Lipocone domain (*Figure 2*, *Figure 1—figure supplement 2*). Of these, a set of three positions, one mapping to each of H2, H3, and H4, can be inferred as being ancestrally present and were likely occupied by a histidine (H2), glutamate (H3), and aspartate (H4), though in some families their identities have secondarily changed (*Figure 2*, *Figure 1—figure supplement 2*). A fourth well-conserved polar position is observed at or near the end of H3; while its ancestral identity is difficult to establish, it is frequently an aspartate or glutamate (*Figure 2*). Two further well-conserved positions are often seen in H4: a polar position downstream of the broadly conserved aspartate residue and a glycine residue near the C-terminus of H4 (*Figure 2*) that likely caps the said helix. Although the ancestral pattern is noticeably degraded in the metazoan Wnt (Met-Wnt) family, it is strongly preserved in the prokaryotic Min-Wnt family (*Figure 1A*). In experimentally determined and modeled structures, the above set of 4 conserved positions forms a predicted active site in the aperture of the Lipocone domain. This, in turn, implies a shared biochemistry across the superfamily, with secondary inactivation in some families like Met-Wnt (see below, *Figure 2*). At the same time, the differences in the specific residues in the conserved positions between different families point to a range of distinct but related activities across the superfamily (*Bastard et al., 2014*; *Glasner et al., 2006*; *Zhang et al., 2014*).

Consistent with these observations, two of the families with intact active sites, the PTDSS1/2 (*Stone and Vance, 1999*; *Tomohiro et al., 2009*) and TelC (*Whitney et al., 2017*), which we identified in this work as members of the Lipocone superfamily, have been characterized as active enzymes operating on different lipid substrates (*Figure 3A*). The eukaryotic PTDSS1/2 localizes to the endoplasmic reticulum (ER) membrane and catalyzes a reaction on the polar head group of phosphatidylcholine or phosphatidylethanolamine (*Saito et al., 1996*; *Stone and Vance, 2000*; *Miyata and Kuge, 2021*). PTDSS1 and PTDSS2, respectively, exchange the phosphate-linked choline or ethanolamine head groups with L-serine (*Stone and Vance, 1999*; *Figure 3A*). The toxin domain of TelC acts on lipid II (*Whitney et al., 2017*), the final intermediate in peptidoglycan biosynthesis, which couples an undecaprenyl diphosphate tail to a head group comprised of a N-acetylmuramic acid-N-acetylglucosamine disaccharide, with a pentapeptide further linked to the former sugar (*Anderson et al., 1967*; *Higashi et al., 1967*). TelC cleaves the bond between the undecaprenol and the diphosphate coupled to the head group (*Whitney et al., 2017*; *Figure 3B*). The reaction is comparable to that catalyzed by PTDSS1/2, as both attack phosphate linkages in lipid head groups. However, TelC apparently directs a water molecule for the attack in lieu of the hydroxyl group of serine directed by PTDSS1/2 (*Figure 3B*).

Combining the above observations, we infer the unified biochemistry for the catalytically active families thus: (1) They act on the head groups of lipids either by removing or swapping phosphate-linked head groups (*Figure 3A–B*). These would be comparable to the phospholipase D (PLD), transphosphatidylation, or polyisoprenol phosphoesterase reactions (*English, 1996*). (2) Given the cone-like cavity and the hydrophobicity of the helices, the lipid tail is predicted to be housed within the lipocone with the head group positioned in the active site. (3) In the case of the integral membrane versions, their orientation would predict the targeting of the head groups of the outer leaf of the bilayer.

## Major clades of the Lipocone superfamily

The extreme sequence divergence of the superfamily, coupled with the small size of the domain, prevents the use of simple phylogenetic tree analyses to resolve its deep evolutionary history. Hence,

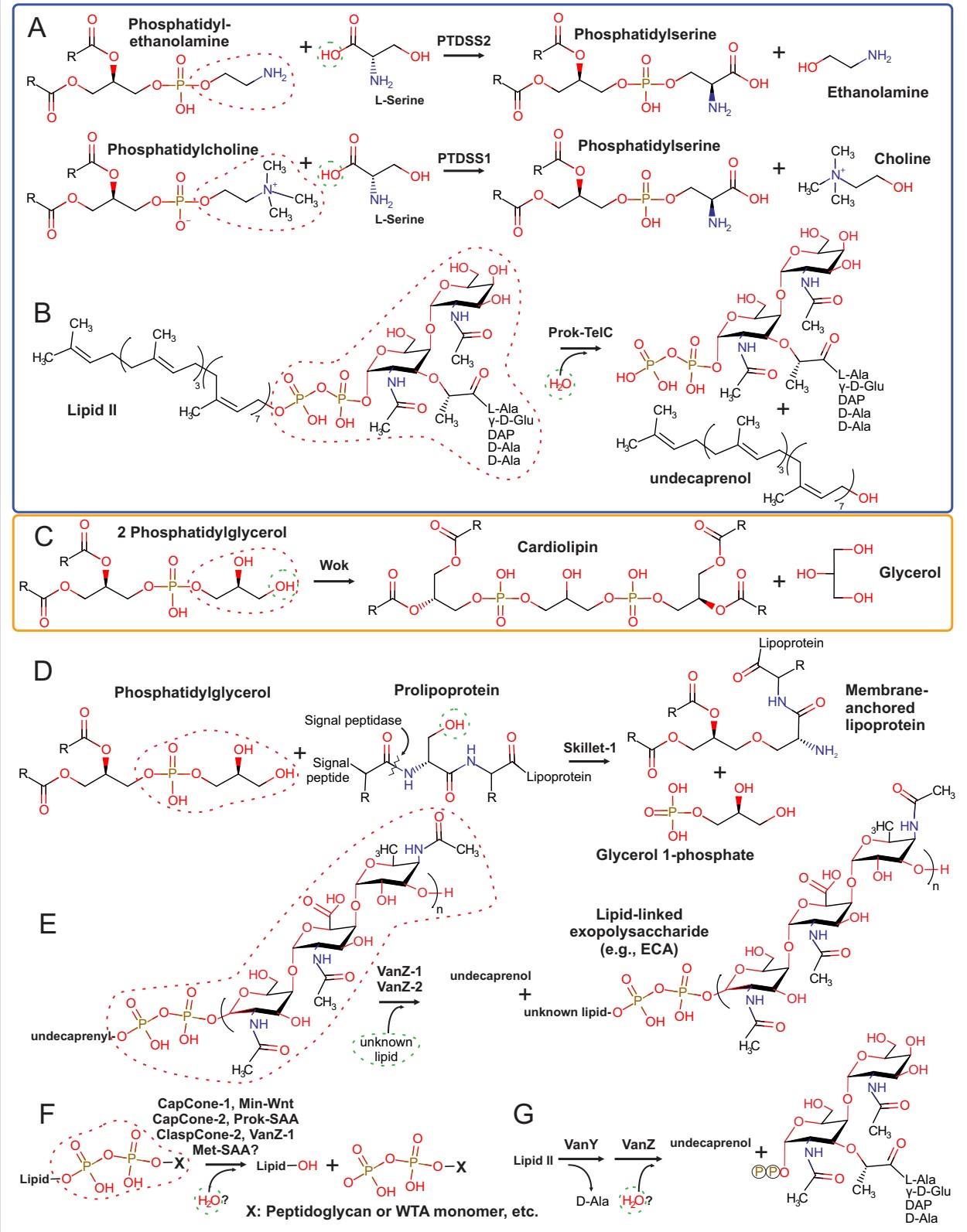

**Figure 3.** Known and predicted Lipocone reaction mechanisms. Experimentally supported reactions are boxed in blue (**A–B**), while a predicted reaction based on genome displacement by a Lipocone domain of an experimentally characterized enzyme is boxed in orange (**C**). The remaining reactions (**D–G**) are suggested based on the contextual inferences in this work. Attacking and leaving groups are denoted by dashed green and red circles, respectively.

we combined community finding algorithms applied on profile-profile similarity networks, comparison of structural features and motifs, and phyletic patterns (*Figures 1B and 2*, *Figure 1—figure supplement 1*) to reconstruct the most parsimonious evolutionary scenario for the diversification of the Lipocone superfamily (*Figure 4*, see Methods). In the below sections, we survey the higher-order clades, highlighting their specific features.

## SAW (SAA-Wnt) clade

This clade consists of four families, with the two prokaryotic families (Min-Wnt and prok-SAA) (*Burroughs and Aravind, 2020*; *Zámocký and Ferianc, 2023*), respectively, giving rise to their counterpart eukaryotic families (Met-Wnt and Met-SAA; *Figures 1A and 4*). This clade is structurally unified by the presence of a fifth helix that stacks in the space between the H2 and H4 helices (*Figure 1A*, *Figure 1—figure supplement 2*). In the Wnt families, this helix is comparable in length to the core helices, while in the SAA families, it is usually shorter (*Figure 1A*). The clade is further unified by the pronounced conservation of a sNxxGR motif (where 's' is a small residue) encompassing the conserved active site position in H4 (*Figure 2*). SAW clade Lipocones show low overall hydrophobicity and are known or predicted to be soluble domains. Outside of the clearly inactive eukaryotic Wnt family, the remaining three families largely conserve the core active site residues (*Figure 2*).

## VanZ-Skillet clade

This clade unites seven families: the two VanZ families, VanZ-1 and VanZ-2, prototyped by the bacterial VanZ protein originally identified in the context of vancomycin resistance, and the five Skillet families, which form a distinct subclade. These are unified by a 'handle'-like structure (hence, 'Skillet'), adopting a helical conformation in the H1-H2 linker (*Figure 1A*, *Figure 1—figure supplement 2*). Strikingly, a symmetric helical handle is present in the H3-H4 loop of the Skillet-DUF2809 and Skillet-3 families (*Figure 1—figure supplement 2*) of this clade. VanZ-1 features a conserved asparagine residue in the H2 position and a DxDDxxxN motif in H4, while VanZ-2 features RKxxH and DxxxD motifs in these respective positions (*Figure 2*). The Skillet families are largely unifiable in their conservation of an ExxQ motif in H3, an aspartate three positions upstream of the canonical H4 aspartate, and another aspartate in the H2 contributing to the active site. These first two features specifically ally them with the VanZ-1 family (*Figure 2*).

While the VanZ domain was previously reported as including a fifth TM helix, which is C-terminal to the 4-helix Lipocone core defined here (*Woods et al., 2018*; *Sur et al., 2021*), our survey instead reveals a striking diversity of configurations around the core 4-helix Lipocone domain (*Figure 5—source data 2*). These range from standalone Lipocone configurations to one or more TM-helices adorning the domain at its N- and/or C-terminus. This variation is consistent with a further tendency for the VanZ families to feature an extensive diversity of domain fusions to both soluble globular domains and discrete TM modules (see below).

The VanZ families are deep-branching, as suggested by their wide phyletic spread (*Figure 1—figure supplement 1*). VanZ-2 is the most widespread individual Lipocone family in bacteria, with several genomes encoding multiple paralogs (*Figure 5—source data 2*; *Woods et al., 2018*; *Stogios and Savchenko, 2020*). It is also found in certain eukaryotes, including a pan-fungal presence and in some representatives of the SAR clade. Both VanZ-1 and VanZ-2 are particularly well-represented in Gram-positive bacterial lineages like Actinomycetota and Firmicutes, while VanZ-2 is nearly universally conserved in the Bacteriodetes/Chlorobi lineage (*Figure 1—figure supplement 1*). In contrast, only one of the Skillet families, Skillet-DUF2809, is widely but sporadically distributed, with the four others being more restricted (*Figure 4*, *Figure 1—figure supplement 1*).

## YfiM clade

This clade includes three families that are consistently centrally located in the profile-profile similarity network (*Figure 1B*). This is likely due to their being close in sequence conservation to the ancestral state of the superfamily (*Figure 2*). Consistent with this, the YfiM-1 family also presents a structurally minimal Lipocone domain, comprised of just the 4-helix configuration with no further elaborations. Notably, this also extends to a lack of domain fusions in this family. In contrast, YfiM-DUF2279 and YfiM-Griddle (DUF3943) are structurally distinguished by an unusual H1-H2 linker (*Figure 1A*), which wraps around the outside and stacks against the H3-H4 linker (*Figure 1—figure supplement 2*). The

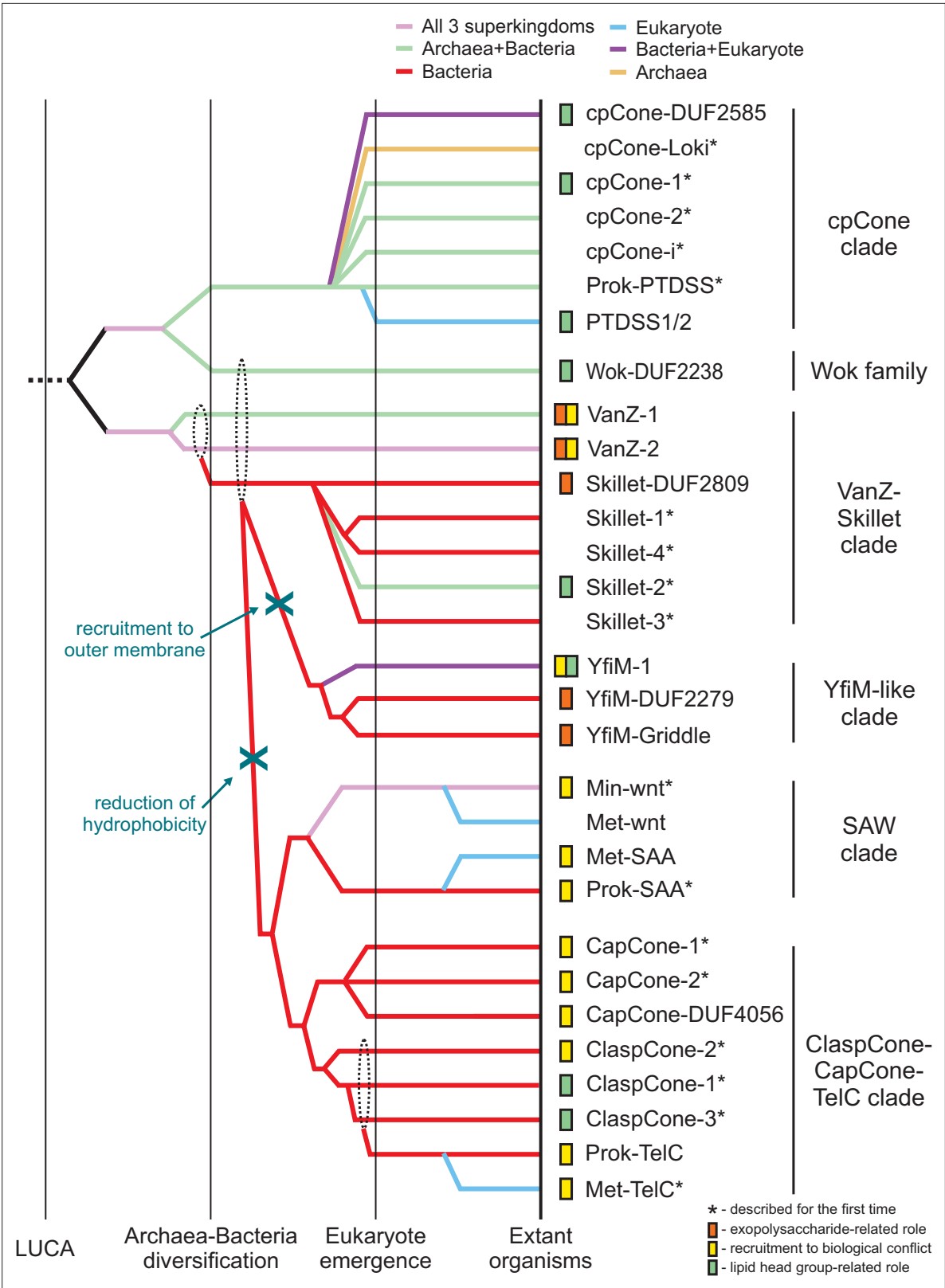

**Figure 4.** Reconstructed evolutionary scenario for the Lipocone superfamily. The relative temporal epochs are demarcated by vertical lines and labeled at the bottom. The clades are represented by colored lines indicating the maximum depth to which the families listed to the right can be traced. Colors track the superkingdom-level phyletic distribution of the family. Dashed-line circles indicate uncertainty in the origin of lineage(s). Inferred or experimentally characterized functions for families are indicated to the left of family names. Asterisks denote newly described families.

YfiM-Griddle family further features a unique 'flattened' surface around the aperture of the Lipocone formed by protruding loops (hence, 'Griddle;' *Figure 1—figure supplement 2*). This leaves the active site pocket more accessible relative to families with more elaborately structured inter-helix linkers. The Griddle family also features a C-terminal extension with a two-helix hairpin (with a hhsP motif in the turn between the two helices, where 'h' is a hydrophobic residue and 's' is a small residue) (*Figure 1—figure supplement 2*). The three YfiM families straddle the membrane-propensity boundary in the plot (*Figure 1C*). Furthermore, the YfiM-DUF2279 and Griddle families are strikingly absent in Gram-positive bacterial lineages (*Figure 1—figure supplement 1*). Concurrent with these features, they are often predicted by the deep-learning-based localization predictor deepTMHMM as outer-membrane proteins, suggesting a role in this subcellular location (see below).

## ClaspCone-CapCone-TelC clade

Members of this clade are unified by an elaborated H1-H2 linker that often contains one or more helical segments that are typically predicted to guard the aperture of the Lipocone domain (*Figure 1—figure supplement 2*). This linker ends in a 'clasp'-like element, which forms a range of structures in different families of the clade before leading into H2 (*Figure 1—figure supplement 2*). The clade is also unified by a striking reduction of overall hydrophobicity, predicting that the members of this clade are soluble domains (*Figure 1C*). Outside of the divergent TelC subclade, most of the families in this clade conserve a serine residue three positions upstream of the active site aspartate in H4, often preceded by an aromatic residue, which is typically phenylalanine. H4 also usually features a conserved asparagine four positions downstream of the conserved aspartate active site position, immediately preceded by a small residue (*Figure 2*). The second H3 active site position is generally poorly conserved, though when present, it is usually an aspartate residue. Finally, H2 contains either a DK or xD motif four positions upstream of the canonical H2 active site histidine residue (*Figure 2*).

The most rudimentary clasps are found in the ClaspCone-1,–2, and –3 families, where it is little more than a rounded loop, though, in ClaspCone-1, a small β-hairpin emerges within it. The three ClaspCone families are further unified by the presence of a two-helix insert leading into H2 that stacks against the Lipocone core (*Figure 1—figure supplement 2*). The three CapCone families, CapCone-DUF4056, CapCone-1, and CapCone-2, are named so for an encasing structure over the active site resembling a cap (*Figure 1—figure supplement 2*). They share a conserved glycine residue six positions upstream of the active site H2 histidine and a S/GxxSxx motif upstream of the conserved H4 aspartate (*Figure 2*). They are further unified by a pronounced β-hairpin clasp augmented by an additional strand (*Figure 1—figure supplement 2*). They also display varying degrees of degeneration of H1, along with family-specific structural elaborations.

The TelC group of this clade, prototyped by the streptococcal TelC toxin (*Whitney et al., 2017*), is divided into two families featuring prokaryotic (prok-TelC) (*Whitney et al., 2017*) and metazoan versions (Met-TelC) (*Dziarski and Gupta, 2006a*). Both TelC families feature a 'cap' with contributions from inserts in the H1-H2 and H3-H4 loops (*Figure 1A*, *Figure 1—figure supplement 1*). Unique to these families is the conservation of an aspartate residue located six positions downstream of the canonical active site aspartate of H4 (*Figure 2*). This aspartate points away from the center of the Lipocone and interacts with a conserved arginine from a synapomorphic C-terminal helical extension.

## cpCone clade

A widespread yet sporadically distributed clade of seven families emerging as a stable community in the profile-profile similarity network (*Figure 1B*) is defined by a unique structural synapomorphy: a circular permutation (*Bliven and Prlić, 2012*) (hence, cpCone) placing the normally N-terminal H1 at the C-terminus of H4 (*Figure 1—figure supplements 2 and 4*). This clade is also united by unique sequence features, viz., a polar residue (typically aspartate) six positions upstream of the conserved H2 histidine and a second glutamate three positions downstream of the conserved H3 glutamate (*Figure 2*). While the circular permutation is shared across the clade, several structural variations are seen, often within the same family (*Figure 1—figure supplement 4*). These include: (1) versions containing a duplication of the Lipocone domain. While the second copy in these versions is catalytically inactive, the H1' from the second duplicate displaces the H1 from the first copy, suggestive of an intermediate to the circular permutation. (2) Versions retaining a candidate H1 that has been displaced by H1' in a five-helix arrangement. (3) Those containing just the circularly permuted core.

(4) Versions showing a degradation of the H1 helix, preserving just a 3-helix core (*Figure 1—figure supplement 4*). Despite this propensity for structural variation, the active site residues are strongly conserved, with the exception of the cpCONE-i family, which we infer to be catalytically inactive (*Figure 2*). The core helices of the cpCone clade are strongly hydrophobic, and they are all predicted to be integral membrane domains (*Figure 1C*). Consistent with this, the eukaryotic PTSSD1/2 domains reside in the ER membrane (*Saito et al., 1996*; *Stone and Vance, 2000*).

## Wok family

The Wok family (partly covered by the Pfam DUF2238 model) shows a higher order grouping with the above circularly permuted clade (*Figures 1B and 4*) but has a phyletic distribution only rivaled by the VanZ-2 family (*Figure 1—figure supplement 1*), suggesting a deep-branching origin. The shape of this family is reminiscent of a wok formed by two distinguishing structural synapomorphies: a 2-TM helix N-terminal extension and a unique 'handle' formed by the linker between the H3 and H4 (*Figure 1—figure supplement 2*). It additionally features a C-terminal, rapidly diversifying cytoplasmic tail. Despite these elaborations, it retains the inferred ancestral active site configuration (*Figure 2*). The strongly hydrophobic core helices of the Wok family predict it to be an integral membrane enzyme (*Figure 1C*).

## Functional themes in the Lipocone superfamily

Given our inference of shared general biochemistry across the Lipocone superfamily in targeting phosphate-containing linkages in head groups of both classic phospholipids and polyisoprenoid lipids, we next used contextual information from conserved gene-neighborhoods, domain architectures and phyletic pattern vectors, a powerful means of deciphering gene function (*Aravind, 2000*), to narrow down the predictions for specific families (*Figure 5*, *Figure 5—source data 1* and *Figure 5—source data 2*). To this end, we constructed a graph (network) wherein the nodes are individual domains and edges indicate adjacency in domain architectures or conserved gene-neighborhoods (*Figure 6*, see Methods). We then identified cliques in these networks and merged the individual cliques containing a particular Lipocone domain to define its dense subgraph (*Figure 6—figure supplements 1–3*). We then analyzed these subgraphs to identify statistically significant functional categories represented in them (*Figure 6—source data 1*; see Methods). This data was combined with existing experimental results and the sequence and structure analyses outlined above to arrive at the functional themes surveyed in the below sections.

## Lipocone domains in membrane lipid, peptidoglycan, and exopolysaccharide modifications

Across different Lipocone families, we found statistically significant connections to roles in modifying lipid head groups in various membranes and in lipids involved in the synthesis of extracellular matrix polymers such as peptidoglycan and lipopolysaccharides (*Figure 6*, *Figure 6—source data 1*, *Figure 5—source data 2*).

### Archetypal lipid head group exchange reactions catalyzed by the cpCone clade

One of the few experimentally characterized Lipocone families is the eukaryotic PTDSS1/2 family of the cpCone clade, members of which exchange the head group of essential membrane phospholipids to generate phosphatidylserine from phosphatidylethanolamine or phosphatidylcholine (*Figure 3A*; *Stone and Vance, 1999*; *Vance, 2018*). Given the pervasive presence of this clade in archaea (*Figure 1—figure supplement 1*), it is thus tempting to speculate that these archaeal cpCones may play a role in the modification of Archaea-specific lipids (*Koga and Morii, 2005*; *Caforio and Driessen, 2017*; *Řezanka et al., 2023*) through a comparable head group exchange reaction (see below).

In bacteria, the related cpCone-1 family shows operonic association with a LolA-like lipoprotein which shuttles lipoproteins to the outer membrane (*Narita and Tokuda, 2010*) and a novel 4TM protein (*Figure 5A*). This raises the possibility that cpCone-1 might mediate the formation of membrane domains featuring lipids with a modified head group that act as foci for the trafficking of lipoproteins. Curiously, the cpCone-1 gene might also be inserted between the bacterial chromosome segregation

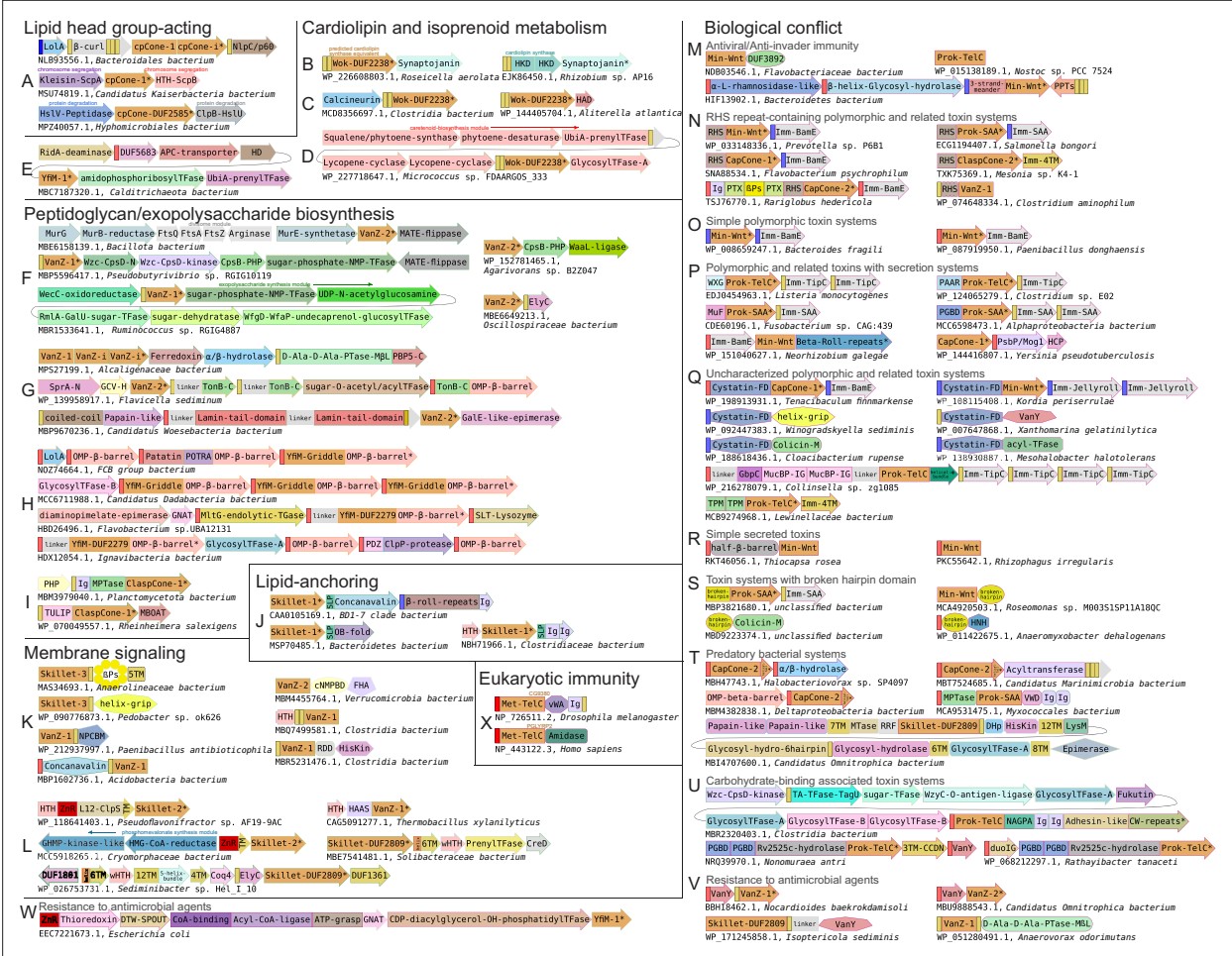

**Figure 5.** Representative contexts for the Lipocone superfamily, grouped by shared functional themes. Genes are depicted by box arrows, with the arrowhead indicating the 3' end of genes. Genes encoding proteins with multiple domains are broken into labeled sections corresponding to them. Domain architectures are depicted by the individual domains represented by distinct shapes. TM segments, lipoboxes (LPs), and signal peptides (SPs) are depicted as unlabeled, narrow yellow, blue, and red rectangles, respectively. All Lipocone domains are consistently colored in orange. Genes marked with asterisks are labeled by the GenBank accession number below each context. Colored labels above genes denote well-known gene names or gene cluster modules. Abbreviations: PTase, peptidase; TFase, transferase; GlycolTFase, Glycosyltransferase; MPTase, metallopeptidase; TGase, transglycosylase; SLP, serine-containing lipobox; cNMPBD, cNMP-binding domain; NCPBM, novel putative carbohydrate binding module; (w) HTH, (winged) helix-turn-helix; ZnR, Zinc ribbon; PPTs, pentapeptide repeats; Imm, immunity protein; βPs, β-propeller repeats; Cystatin-FD, Cystatin fold domain; MTase, methylase; PGBD, peptidoglycan-binding domain; MβL, metallo-β-lactamase; L12-ClpS, ClpS-ribosomal L7/L12 domain; TA, teichoic acid.

The online version of this article includes the following source data and figure supplement(s) for figure 5:

**Source data 1.** Table of Lipocone family conserved contextual associations across distinct functional themes.

**Source data 2.** List of identified genome contexts.

**Figure supplement 1.** Multiple sequence alignment of serine-containing lipobox (SLP).

**Figure supplement 2.** Structural overview of newly identified immunity proteins pairing with toxin-containing proteins in polymorphic and allied toxin systems.

**Figure supplement 3.** Sequence and structure overview of the broken-hairpin domain.

and condensation complex subunits, the Kleisin ScpA and the wHTH ScpB (***Kamada et al., 2013***; ***Schleiffer et al., 2003***; ***Soppa et al., 2002***; ***Aravind et al., 2005***). The bacterial cpCONE-DUF2585 is operonically coupled to a GNAT family NH₂-group-acetyltransferase and further linked to genes for the glycolate oxidase GlcE and GlcF (***Pellicer et al., 1996***) and the bacterial proteasome subunits HslV and HslU (***Ramachandran et al., 2002***; ***Figure 5A***, ***Figure 5—source data 2***). These might point to the coupling of membrane lipid head group modifications with disparate processes, such

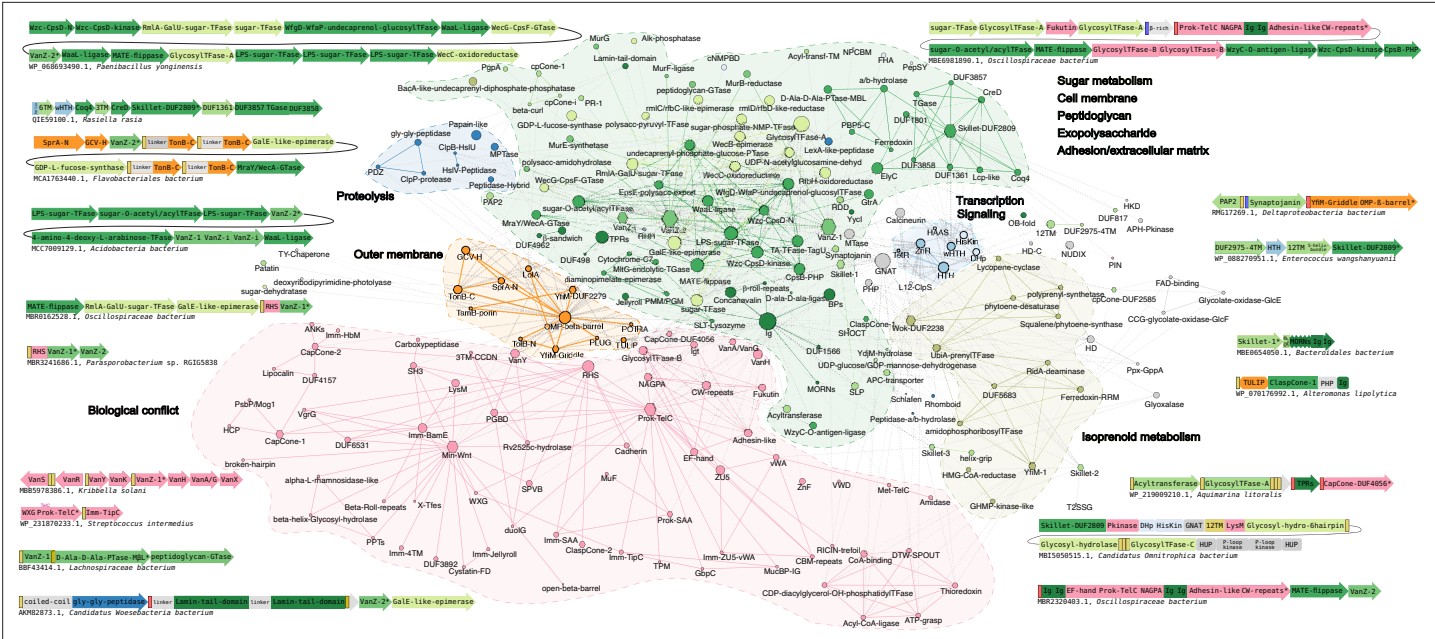

**Figure 6.** Lipocone contextual network. The network represents the conserved contextual associations of Lipocone domains (hexagonal nodes). Nodes and edges are colored based on known or inferred functional categories of the domains. The nodes are scaled by their degree. Gray coloring indicates domains without specific functional assignments. Examples of conserved gene neighborhoods and domain architectures supplementing those in *Figure 5* illustrate contexts that bridge functional themes. Here, individual domains are colored to match network coloring. Additional abbreviations to those in *Figure 5*: APH-Pkinase, aminoglycoside phosphotransferase-like kinase; HUP, HIGH, UspA and PP-ATPase superfamily-like domain; Alk-phosphatase, Alkaline phosphatase; dehyd, dehydrogenase; TPRs, tetratricopeptide repeats; PMM/PGM, phosphomannomutase/phosphoglucomutase; ZnF, zinc finger; APC-transporter, amino acid-polyamine-organocation transporter; LPS, lipopolysaccharide.

The online version of this article includes the following source data and figure supplement(s) for figure 6:

**Source data 1.** Significant enrichment of Lipocone family contextual associations across functional categories.

**Source data 2.** *Figure 6* network and node annotation YAML files.

**Figure supplement 1.** Lipocone domain-centered subgraphs of contextual network in *Figure 6*.

**Figure supplement 2.** Lipocone domain-centered subgraphs of contextual network in *Figure 6*.

**Figure supplement 3.** Lipocone domain-centered subgraphs of contextual network in *Figure 6*.

as chromosome segregation during cell division or different responses to stress (*Storck et al., 2018*; *Barák and Muchová, 2013*; *Hauck and Bernlohr, 2016*).

## The Wok and YfiM-1 families in cardiolipin and modified isoprenoid lipid pathways

We observed a set of conserved gene neighborhoods displaying the mutually exclusive presence of a synaptojanin-like phosphatase gene, with one encoding either a member of the Wok family or a cardiolipin synthase of the HKD superfamily (*Guo and Tropp, 2000*; *Figure 5B*, *Figure 5—source data 2*). This suggested that the latter two are analogous enzymes catalyzing equivalent reactions. The cardiolipin synthase utilizes two phosphatidylglycerol molecules as substrates to generate cardiolipin with the release of one of the glycerol head groups (*Tan et al., 2012*). This is comparable to the head group exchange reaction catalyzed by PTDSS1/2 from the cpCone clade (*Figure 3A*). Hence, we propose that these members of the Wok clade are cardiolipin synthases (*Figure 3C*). Distinct phosphoesterases, namely the synaptojanin-like, calcineurin-like (*Aravind and Koonin, 1998a*) and HAD (*Burroughs et al., 2006*) enzymes, are also observed in gene-neighborhood associations with the Wok, suggesting that they might together regulate membrane lipid composition by acting on the phospholipids or their precursors (*Figure 5C*). In a distinct neighborhood, the Wok clade enzyme is coupled to carotenoid biosynthesis genes (*Vershinin, 1999*; *Šesták et al., 2004*). (*Figure 5D*, *Figure 6—figure supplement 1*). This raises the possibility that these members might also catalyze

a comparable reaction to the above on isoprenoid lipids: for instance, they could synthesize a carotenoid from two geranylgeranyl-diphosphate molecules (*Sandmann and Misawa, 1992*; *Chamovitz et al., 1992*). In both of these contexts, the actinobacterial operons often include genes for GT-A family glycosyltransferases, suggesting the further synthesis of glycosylated derivatives of the lipids or carotenoids (*Liu and Mushegian, 2003*; *Figure 5—source data 2*). In several bacteria, a YfiM-1 family Lipocone is operonically coupled to a UbiA-like prenyltransferase (*Tran and Clarke, 2007*). This gene neighborhood additionally codes for a slew of enzymes, such as an amidophosphoribosyltransferase (*Massière and Badet-Denisot, 1998*), a RidA-like deaminase (*Liu et al., 2016*), and a pair of structurally distinct phosphoesterases, respectively, containing an HD and a PHP domain (*Aravind and Koonin, 1998a*; *Aravind and Koonin, 1998b*; *Figure 5E*, *Figure 6—figure supplement 1*, *Figure 5—source data 2*). This suggests a role for the YfiM-1 Lipocone and the associated enzymes in generating a modified polyisoprenoid metabolite.

## VanZ families modifying lipid head groups in peptidoglycan and exopolysaccharide metabolism

The widespread VanZ-1 and VanZ-2 families (*Figure 1A*) frequently show either gene neighborhood associations or direct domain fusions, with diverse genes involved in both peptidoglycan and other extracellular polysaccharide pathways. Chief among these are the lipid carrier flippase (Pfam: MviN_MATE clan) (*Becker et al., 1993*; *Ruiz, 2015*; *Ruiz, 2008*), the UDP-GlcNAc/MurNAc lipid transferases, which generate the lipid-linked exopolysaccharide precursors (lipid I) (*Higashi et al., 1967*; *Heydanek et al., 1969*), and UDP-N-acetylglucosamine (UDP-GlcNAc) biosynthesis enzymes (*Mengin-Lecreulx and van Heijenoort, 1993*; *Mengin-Lecreulx and van Heijenoort, 1996*). Despite certain examples of crossover in functional themes, the gene-neighborhood contexts of VanZ-1 and VanZ-2 suggest a metabolic partitioning, with VanZ-2 significantly associating specifically with peptidoglycan-related genes and VanZ-1 significantly linking with biosynthesis genes for other exopolysaccharides (e.g. the outer-membrane-associated lipopolysaccharide) (*van Heijenoort, 2007*; *Figures 5F and 6*, *Figure 6—figure supplement 2*, *Figure 6—source data 1*). The latter includes WaaL-like lipid A transferase (*Ashraf et al., 2022*), the polysaccharide chain-length determination domain Wzz (*Franco et al., 1998*), the Wzc kinase and the 'extracellular antigen'-regulating ElyC-like domain (Pfam: DUF218) (*Rai et al., 2021*), and numerous nucleotide-diphosphate sugar biosynthesis and modification enzymes (*Suresh Kumar et al., 2007*; *Figures 5F and 6*, *Figure 6—figure supplement 2*).

The precursors of both peptidoglycan and exopolysaccharides are synthesized in the cytosol, linked to lipid carriers via a diphosphate linkage, e.g., the polyisoprenoid lipid undecaprenol (bactoprenol) (*van Heijenoort, 2007*; *Suresh Kumar et al., 2007*; *van Heijenoort, 2001*; *Hong et al., 2023*; *Rai and Mitchell, 2020*). A key step in their maturation is the flipping by the flippase of the lipid-linked intermediates associated with the inner membrane to the outer membrane. These flipped units are then incorporated into the maturing chain (*Sham et al., 2014*; *Kim et al., 2018*) by the peptidoglycan glycosyltransferase (GTase) (*Di Guilmi et al., 2003*) and the chain length determination protein, WzzE/polymerase (WzyE) (*Franco et al., 1998*; *Weckener et al., 2023*), in peptidoglycan and other exopolysaccharide maturation pathways, respectively. Based on the precedence of the TelC-catalyzed reaction (*Figure 3B*), we propose that VanZ-1 and VanZ-2 comparably act on the flipped lipid II head groups bearing the modified sugar intermediates to release the undecaprenol via phosphoester cleavage (*Figure 3F*). Such activity could modulate the concentration of available peptidoglycan intermediates and allow formation of peptidoglycan with varying thickness and composition during different phases of the life cycle, e.g., sporulation versus vegetative growth in Bacillota. Such a reaction could also possibly modulate exopolysaccharide biosynthesis by comparably acting on their precursors.

The terminal transfer from the lipid carrier of the Gram-negative bacterial O-antigen (as well as other exopolysaccharides attached to the lipid A carrier) has been attributed to the WaaL-like enzymes (*Ashraf et al., 2022*; *Han et al., 2012*). However, bacteria generate further lineage-specific polysaccharide decorations, capsule structures, and other exopolysaccharides (e.g. xanthan, enterobacterial common antigen (ECA), alginate, colonic acid), as well as teichoic acids (e.g. wall teichoic acids, WTA) (*Imperiali, 2019*; *Mostowy and Holt, 2018*). Notably, the analogs of WaaL, i.e., the terminal transferases for several exopolysaccharides, including ECA and WTA, have to date escaped identification (*Rai et al., 2021*). Hence, it is possible that, by analogy to the PTDSS1/2 reaction (*Figure 3A*), the

VanZ families act on the lipid carrier-linked sugar head groups to catalyze either the extension of the polysaccharide chains through transesterification or the terminal release of the mature chain through phosphoester cleavage (*Figure 3E*).

## Atypical VanZ domains in uncharacterized modifications of peptidoglycan and the outer membrane

Certain representatives of the two VanZ families also show operonic associations indicative of outer membrane-associated or peptidoglycan modification functions distinct from those described above (*Figures 5G and 6*, *Figure 5—source data 2*): (1) An operon in FCB group bacteria couples a VanZ-2 gene with those coding for a SprA secretin-like channel protein (*Saiki and Konishi, 2007*), a glycine cleavage H (GCVH)-like lipoyl-group carrier protein (*Pares et al., 1994*), a 2TM protein fused via a proline-rich linker to a C-terminal TonB-C domain (*Shultis et al., 2006*), and a secreted, second TonB-C domain fused to a Wzi-like outer membrane protein (OMP) superfamily β-barrel (*Rahn et al., 2003*; *Figures 5G and 6*). (2) In betaproteobacteria, certain VanZ-1 domains are duplicated with the C-terminal copy being inactive (VanZ-i) and found in an unusual four-gene operon with a thioredoxin-fold [2Fe-2S] ferredoxin (*Qi and Grishin, 2005*), a possible lipase of the α/β-hydrolase superfamily (*Suplatov et al., 2012*), and a metallo-β-lactamase (MβL) fold D-Ala-D-Ala cross-linking transpeptidase (*Palomeque-Messia et al., 1991*; *Aravind, 1999*). (3) A patescibacterial operon encodes a VanZ-2 domain with an ABC ATPase transporter system, either of two structurally distinct peptidases, namely a papain-like or glycine-glycine peptidase (*Novinec and Lenarčič, 2013*; *Razew et al., 2022*), fused to the same membrane-anchored N-terminal coiled-coil region, and a further TM protein containing one or more external Lamin-Tail domains (LTDs) predicted to bind extracellular DNA or polysaccharides (*Mans et al., 2004*; *Figures 5G and 6*, *Figure 6—figure supplement 2*). The associations in the first of the above neighborhoods point to a distinct outer membrane-associated lipid modification, while the other two might be involved in lineage-specific decorations/modifications of peptidoglycan, accompanied by peptide-crosslinking or cleavage activities.

## Lipocone domains operating in the outer membrane

Contextual associations, phyletic patterns, and localization predictions support the action of two Lipocone families directly in the outer membrane. Notably, the YfiM-Griddle and YfiM-DUF2279 families are found nearly obligately directly fused or operonically linked to several distinct OMP β-barrels (*Wimley, 2003*; *Fairman et al., 2011*; *Figures 5H and 6*, *Figure 6—figure supplement 1*). Up to three YfiM-Griddle Lipocones, usually with a cognate OMP β-barrel, might be encoded next to each other in the genome. Additionally, YfiM-Griddle family genes are often encoded in operons with several components of the outer membrane lipid and protein trafficking apparatus, including the LolA-like chaperone (*Tokuda and Matsuyama, 2004*), the POTRA domain (*Sánchez-Pulido et al., 2003*; *Kim et al., 2007*), the channel-blocking Plug domains (*Oke et al., 2004*), and the TolA-binding TolB-N domain (*Carr et al., 2000*). Further, these operons might encode a Patatin-like lipase (*Ghosh et al., 2006*), GT-B family glycosyltransferases (*Liu and Mushegian, 2003*), and a range of phosphoesterases (e.g. an integral membrane phosphatidic acid phosphatase PAP2 *Stukey and Carman, 1997*) a lipobox-containing synaptojanin superfamily phosphoesterase (*Whisstock et al., 2000*), and a secreted R-P phosphatase *Burroughs and Aravind, 2023*, see *Figures 5H and 6*, *Figure 5—source data 2*. In addition to the fusion to the OMP β-barrel, the YfiM-DUF2279 family (*Figure 5H*) shows operonic associations with a secreted MltG-like peptidoglycan lytic transglycosylase (*Höltje et al., 1975*; *Yunck et al., 2016*), a lipid-anchored cytochrome c heme-binding domain (*Einsle et al., 1999*), a phosphoglucomutase/phosphomannomutase enzyme (*Levin et al., 1999*), a GNAT acyltransferase (*Dong et al., 2007*), a diaminopimelate (DAP) epimerase (*Antia et al., 1957*), and a lysozyme-like enzyme (*Koraimann, 2003*). In a distinct operon, YfiM-DUF2279 is combined with a GT-A glycosyltransferase domain (*Liu and Mushegian, 2003*), a further OMP β-barrel, and a secreted PDZ-like domain fused to a ClpP-like serine protease (*Muley et al., 2019*; *Hara et al., 1991*; *Figure 5H*).

The strong linkage to the OMP β-barrel, together with their predicted localization, suggests that these YfiM-Griddle and YfiM-DUF2279 Lipocone domains operate in the outer membrane, potentially in concert with both cytoplasmic carbohydrate biosynthetic modules and periplasmic lipid- and carbohydrate-processing enzymes. As with the inner membrane lipids, they could potentially catalyze

modifications of head groups through transesterification and/or linkage/release of outer membrane-associated polysaccharide chains through action on lipid-head group phosphoesters.

## Lipocone domains acting on lipids in transit to the outer membrane

The ClaspCone-1 and ClaspCone-3 families lack the hydrophobicity indicative of direct residence in the membrane (*Figure 1C*); instead, they are predicted to localize to the periplasmic space. In the ClaspCone-1 family, the Lipocone domain is fused at the extreme N-terminus to either a single TM or a 5TM domain predicted to anchor it to the cell membrane. Between this TM element and the Lipocone domain, we detected a previously uncharacterized version of the Tubular lipid binding protein (TULIP) domain (*Wong and Levine, 2017*; *Levine, 2019*) or an Ig-like and a Zincin-like metallopeptidase (MPTase) domain (*Dhanaraj et al., 1996*; *Figures 5I and 6*). These ClaspCone-1 genes may also show operonic associations with genes encoding a lipase of the SGNH family (*Lo et al., 2003*) and a membrane-bound O-acyltransferase (MBOAT; *Figure 5I*, *Figure 5—source data 2*; *Hofmann, 2000*). The TULIP domain superfamily has recently been characterized as a lipid-binding domain (*Wong and Levine, 2017*; *Levine, 2019*), which in proteobacteria functions in outer membrane lipid transport (*Rahlwes et al., 2017*; *Yeow and Chng, 2022*). Thus, we propose that the ClaspCone-1 family is likely to act in the periplasmic space on the head groups of outer-membrane targeted lipids bound to the TULIP or potentially to the Ig-like domains occupying an equivalent position in the domain architecture.

## A Lipocone domain catalyzing a predicted lipoprotein lipid linkage reaction

The Skillet-1 Lipocone is strongly coupled in an operon with a downstream gene coding for a protein with an unusual lipobox-like sequence followed by one of several extracellular domains (e.g., concanavalin, β-jelly roll, OB-fold, Ig-like, β-propeller) predicted to bind carbohydrates or other ligands (*Kadirvelraj et al., 2008*; *Flint et al., 2004*; *Murzin, 1993*; *Williams and Barclay, 1988*; *Chen et al., 2011*; *Figures 5J and 6*, *Figure 6—figure supplement 1*, *Figure 5—source data 2*). The lipobox-like sequence features a conserved GS motif at its C-terminus instead of the usual GC of the classic lipobox of bacterial lipoproteins (*Babu et al., 2006*; *Figure 5—figure supplement 1*). In the canonical lipoprotein processing pathway, a thioether linkage is formed between the sulfhydryl of the cysteine and a diacylglycerol lipid embedded in the inner membrane by the lipoprotein diacylglyceryl transferase (lgt) enzyme, followed by the cleavage of the signal peptide at the GC motif junction by the signal peptidase (*Sankaran and Wu, 1994*; *Tjalsma et al., 1999*). Given the serine in place of the cysteine in these lipobox-like sequences, we propose that it undergoes non-canonical lipidation by the associated Skillet-1 Lipocone protein in lieu of lgt. We propose that, comparable to PTDSS1/2, which act on free serine, the Skillet-1 family links the conserved serine from the lipobox-like sequence to a phospholipid (*Figure 3A and D*).

## Lipocone domains in predicted lipid-associated signaling systems

### Systems defined by standalone proteins with Lipocone domains

Several representatives of the two VanZ and Skillet-3 families are fused to a diverse array of known or predicted extracellular ligand-binding domains (*Figure 5K*), where the architecture takes the form of SP + X + TM + Lipocone or Lipocone + TM + X, where 'X' is the extracellular ligand-binding domain and SP is a signal peptide. The ligand binding domains include: (i) carbohydrate-binding lectin domains such as jelly-roll, concanavalin-like, NPCBM-like, CBD9-like, and other β-sandwiches (*Kadirvelraj et al., 2008*; *Flint et al., 2004*; *Notenboom et al., 2001*; *Rigden, 2005*; *Boraston et al., 2004*) (ii) a lipid-binding helix-grip superfamily domain (*Iyer et al., 2001*) (iii) those binding other potential ligands (e.g. Ig, OB-fold, YycI-like, DUF498-like, PepSY-like, β-helix, TPR, MORN, and β-propeller repeats *Murzin, 1993*; *Williams and Barclay, 1988*; *Chen et al., 2011*; *Boraston et al., 2004*; *Santelli et al., 2007*; *Das et al., 2001*; *Bennett et al., 2013*; *Cortajarena and Regan, 2006*; *Figure 5K*, *Figure 5—source data 2*). We interpret these architectures as implying signaling, wherein the binding of the cognate ligand by one of the above domains regulates the catalytic activity of the associated Lipocone domain. Among these, the extracellular domains fused to the Skillet-3 family are particularly notable for their extreme variability (*Figure 5K*). This suggests diversification under an arms race scenario (also see below) in a biological conflict. Further, the genes coding for the above are

sporadically associated with exopolysaccharide metabolism genes (*Figure 5—source data 2*). Hence, it is conceivable that this signaling is associated with exopolysaccharide variation (e.g., O-antigen phase-variation *Seed et al., 2012*; *Cai et al., 2019*), which might play a role in evading bacteriophage attachment.

Additionally, VanZ-1 Lipocone domains are also fused to several known signaling domains confidently predicted to reside in the cytoplasm, including the cyclic nucleotide-binding domain (cNMPBD), phosphopeptide-binding FHA, and DNA-binding RHH and HTH domains (*Aravind et al., 2005*; *Yau, 1994*; *Durocher et al., 1999*; *Schreiter and Drennan, 2007*; *Figure 5K*). These associations suggest potential VanZ regulation via a cytoplasmic cyclic nucleotide (sensed by cNMPBD) or, conversely, VanZ acting as an allosteric regulator of a transcriptional program via the HTH or RHH domain. One of the most common yet enigmatic fusions to VanZ is with the integral membrane RDD domain (*Stogios and Savchenko, 2020*). The role of this domain is unknown; however, our analysis indicates that it contains a conserved intra-membrane binding site oriented towards the cytoplasmic face of the membrane (Nicastro GN, Burroughs AM, Aravind L, manuscript in preparation). The VanZ-RDD fusion is sometimes further fused to other domains (*Figure 5K*), the most notable being a highly derived but active novel histidine kinase domain (*Figure 5K*). Together, these associations point to the coupling of lipid modification with a signaling event on the cytoplasmic face of the membrane, which might relate to the dynamic regulation of lipid-carrier-bound exopolysaccharide precursors.

## Multi-component associations of the Lipocone proteins in signaling

These systems resemble the above-discussed versions but are encoded by conserved gene neighborhoods that separate the Lipocone and the signaling elements (typically predicted transcription regulators) into distinct genes. Our analysis recovered at least three such systems: (1) A VanZ-1 Lipocone in the recently described HAAS/PadR-HTH two-component systems, which sometimes replace classical Histidine kinase-Receiver two-component systems (*Ravi et al., 2024*). In these systems, the detection of an extracellular or intramembrane stimulus by a sensor domain releases the PadR-HTH transcription regulator bound to the sensor-fused HAAS domain. Here, VanZ-1 occupies the sensor position (*Figure 5L*). (2) A Skillet-2 Lipocone is coupled in a core two-gene system to a conserved upstream gene (*Figure 5L*, *Figure 6—figure supplement 1*). That gene encodes a single TM protein with either a zinc ribbon (ZnR) fused to a conserved helix or an HTH domain fused to a ClpS-ribosomal L7/L12 domain in its cytoplasmic region (*AhYoung et al., 2016*). These neighborhoods might also code for an HMG-CoA reductase and GHMP kinase that catalyze successive reactions in the production of phosphomevalonate, a precursor of isoprenoid lipids (*Durr and Rudney, 1960*; *Tchen, 1958*). (3) A Skillet-DUF2809 Lipocone protein is operonically coupled with a 6TM protein and a further predicted transcription factor with a wHTH protein. These operons are further elaborated via additional embedded and flanking genes, either coding for components of isoprenoid lipid (e.g. undecaprenol) (*Wolff et al., 2003*; *Guo et al., 2005*) or exopolysaccharide (e.g. ECA and related polysaccharides) metabolism (*Suresh Kumar et al., 2007*; *Rai and Mitchell, 2020*; *Figures 5L and 6*, *Figure 6—figure supplement 2*, *Figure 5—source data 2*).

The Lipocone domains in these systems are predicted to be active enzymes, which, together with their operonic associations, point to functions involving the modification or transesterification of isoprenoid lipid head groups, sometimes in the context of exopolysaccharide biosynthesis. However, their associations with the intracellular HTH domains suggest that the Lipocone enzymatic activity is potentially coupled with the transcriptional regulation of the production of precursors of the lipids or exopolysaccharides. Given the high variability in the associated genes related to exopolysaccharide/lipopolysaccharide biosynthesis, we anticipate that the associated transcriptional regulation potentially relates to functional categories showing high diversity across bacteria, such as responses to environmental stress, phages, predatory bacteria attacks, or host immune response.

## Lipocone domains as effectors in biological conflicts
### Lipocone domains in antiviral immunity

The Min-Wnt domains (*Figure 1A*) that we originally identified were predicted to play a role in biological conflicts with invasive selfish elements, such as viruses (*Burroughs and Aravind, 2020*). In this work, we better explain their potential mechanism of action. These versions show no fusions to extracellular domains or secretory signals, suggesting that they are deployed from within the bacterial cell

(*Figure 1C*). These Min-Wnts are typically fused to the DUF3892, which displays a fold characterized by a three-stranded meander followed by a helix also seen in the dsRNA-binding domain and the ribosome hibernation factors (HPF) (*Burroughs and Aravind, 2016*; *Ueta et al., 2008*; *Figure 5M*). Hence, we propose that these versions might potentially act to sense virally induced RNAs or modified ribosomes (*Burroughs and Aravind, 2020*) to trigger a dormancy or suicide response to limit viral infection via the Min-Wnt effector. Specifically, the Min-Wnt might attack peptidoglycan precursors, such as lipid II, prior to their 'flipping' to restrict cell wall synthesis (*Ruiz, 2015*; *Akanuma et al., 2016*; *Kuk et al., 2022*) or other such carrier lipids.

One other Min-Wnt domain, N-terminally fused to a three-stranded β-meander, is pervasive in the Bacteroidetes clade. This is operonically coupled with genes encoding a TM-linked run of penta-peptide repeats and two structurally distinct, secreted glycosyl hydrolase enzymes, respectively, containing a TIM barrel domain and a run of β-helix repeats (*Figure 5M*, *Figure 5—source data 2*). Furthermore, cyanobacteria show a standalone prok-TelC domain without any secretory signals. These could again act as effectors targeting lipid-linked precursors of peptidoglycan or exopolysaccharides in response to intracellular invaders or stress (*Figure 5M*). Interestingly, some tailed bacteriophages also code for intracellular Min-Wnt domains, suggesting that they might also be deployed on the virus side in biological conflicts, such as limiting superinfection (*Figure 5—source data 2*).

## Lipocones as toxin domains in polymorphic and allied conflict systems

Polymorphic toxins and related systems, widespread across bacteria and certain archaea, are characterized by a highly variable C-terminal toxin domain ('toxin tip') that is preceded by a range of more conserved domains typically required for autoproteolytic processing of the toxin, its packaging and trafficking (e.g. RHS repeats), adhesion and secretion via one of several secretory systems (*Zhang et al., 2012*; *Iyer et al., 2011*). The toxin might be delivered via one of the secretory systems into a target cell or else via direct contact between interacting cells. Classical polymorphic toxins are usually involved in kin discrimination and are accompanied by genomically linked cognate immunity proteins that protect against self-intoxication (*Zhang et al., 2012*; *Ruhe et al., 2020*). Keeping with the principle of effector sharing between systems involved in distinct types of biological conflicts, we had originally identified a Min-Wnt domain closely related to those described in the above subsection as a toxin tip in polymorphic toxin systems (*Burroughs and Aravind, 2020*). In the current work, we extend these findings to show that several distinct Lipocone families have been independently recruited as toxin tips of polymorphic toxins and related systems, namely Min-Wnt, prok-SAA, prok-TelC, CapCone-1, CapCone-2, ClaspCone-2, and VanZ-1 (*Figures 4 and 5N*, *Figure 6 Figure 6—figure supplement 3*).

Certain CapCone-2 and Min-Wnt toxins from Gram-positive bacteria define some of the simplest of these toxin systems. Here, a standalone Lipocone domain is coupled to a signal peptide or lipobox via a poorly structured linker. These are usually encoded in a two-gene configuration with their cognate immunity protein (*Figure 5O*). More complex versions present, in addition to adhesion, peptidoglycan-binding, lipid-binding and proteolytic processing domains, multiple hallmarks of delivery through specific secretion systems. These include T4SS (VirD4-binding domain), T6SS (PAAR domain), T7SS (WXG/LXG domain), T9SS, and MuF domains (*Zhang et al., 2012*; *Ruhe et al., 2020*; *Figure 5P*, *Figure 5—source data 2*). Additionally, we recovered standalone CapCone-1 domains encoded in an operon with a PsbP/MOG1 superfamily domain diagnostic of secretion via the T6SS (*Zhang et al., 2012*; *Alcoforado Diniz and Coulthurst, 2015*; *Figure 5P*). Further, we also found Min-Wnt domains fused to the N-termini of RTX-like β-roll repeats, suggestive of T1SS-mediated export (*Satchell, 2011*; *Figure 5P*).

Our analysis also uncovered multiple, previously uncharacterized trafficking/packaging systems associated with different Lipocone polymorphic toxins. Several Min-Wnt and CapCone-1 domains with lipoboxes are fused to an N-terminal Cystatin-like superfamily domain (*Kordis and Turk, 2009*; *Figure 5Q*). The same domain is also comparably fused to several other C-terminal toxin domains in related organisms, some of which are also predicted to target lipid head groups: (i) a novel toxin domain we unified with the lipid-targeting Colicin M fold (*Chérier et al., 2021*) (ii) a lipid-binding START-domain-like helix-grip fold domain (*Iyer et al., 2001*) (iii) a papain-like fold fatty acyltransferase (*Anantharaman and Aravind, 2003*) (iv) a domain related to the VanY-like D-Ala-D-Ala carboxypeptidase (*Arthur et al., 1992*; *Figure 5Q*, *Figure 5—source data 2*). In all these cases, the toxins are

coupled to a related immunity protein (see below), suggesting that they define a distinct polymorphic toxin system. We propose that this Cystatin-like domain specifies a novel packaging or deployment system upon secretion for the C-terminal toxin domain, analogous to Cystatin domains in functioning with eukaryotic proteases (*Turk and Bode, 1991*). The prok-TelC family Lipocones are found in distinctive architectures in two poorly characterized, predicted polymorphic toxin systems. In one of them, they are fused to an N-terminal glucan-binding GbpC β-sandwich domain (*Sato et al., 1997*) and repeats of MucBP-like Ig domains (*MacKenzie et al., 2009*), which might anchor them to exopolysaccharides (*Figure 5Q*, *Figure 6—figure supplement 3*, *Figure 5—source data 2*). The second variant found in association with T9SS components (*Chagnot et al., 2013*) shows fusions to one or more copies of a previously undetected TPM domain (*Figure 5Q*). While the domain has been claimed to be a phosphatase (*Wu et al., 2011*), our recent analysis indicates that this is unlikely to be the case (*Ravi et al., 2024*). Instead, we propose that the TPM domain might assist in assembling membrane-linked protein complexes, a role that might be relevant to the trafficking of these toxins (*Ravi et al., 2024*).

To date, the only experimentally characterized Lipocone domain from polymorphic toxins is of the prok-TelC family that are secreted via T7SS (*Whitney et al., 2017*; *Cao et al., 2016*; *Figures 5P and 6*). Notably, prok-TelC has been shown to be active only outside the cell and not in the cytoplasm (*Whitney et al., 2017*). As noted above, it attacks lipid II to cleave off the peptide-linked disaccharide pyrophosphate head group from the undecaprenol tail (*Figure 3B*). Prok-TelC has also been speculated to similarly attack WTA-lipid II linkages (*Whitney et al., 2017*). These findings provide a template for other Lipocone superfamily effectors in potentially targeting lipid carrier linkages in peptidoglycan and exopolysaccharide intermediates. However, given the diversity within the family (*Figure 3F*), it is conceivable that they also target other lipids.

## Immunity proteins of Lipocone polymorphic toxins indicate periplasmic/ intramembrane action

To date, only a single immunity protein has been reported for Lipocone toxins, viz., TipC, which counters prok-TelC toxin in the periplasm (*Whitney et al., 2017*; *Klein et al., 2018*; *Figures 5P and 6*). Here, we uncovered a range of immunity proteins belonging to structurally distinct folds that counter the remaining Lipocone toxins (*Figure 5N–Q*, *Figure 5—figure supplement 2*, *Figure 5—source data 2*). The most widespread of these is a rapidly evolving, membrane-anchored member of the BamE-like superfamily that associates with not only Min-Wnt and CapCone toxins but also other above-mentioned lipid-head-group targeting toxins (e.g. the novel Colicin M-like domain). The BamE-like fold features a core two-helix hairpin followed by a run of three β-strands (*Figure 5—figure supplement 2*). The classical BamE operates in a pathway for the assembly of OMP β-barrels (*Knowles et al., 2011*; *Hagan and Kahne, 2011*), suggesting that these immunity proteins emerged from an ancestral BamE and, like it, function in the periplasm. Additional candidate immunity proteins with more restricted phyletic spreads include (*Figure 5—figure supplement 2*): (i) a β-jelly-roll fold-containing protein (*Flint et al., 2004*) (ii) an integral membrane protein with a 4-TM core. These two are observed with Min-Wnt toxins. (iii) A novel domain combining an α-helix with a run of 4 β-strands stabilized by four absolutely conserved cysteine residues. This is coupled to both Min-Wnt and prok-SAA toxins; (iv) a protein with an OB-fold domain (*Murzin, 1993*) (v) a protein with a β-sandwich related to the eukaryotic centriolar assembly SAS-6 N-terminal domain (*Kitagawa et al., 2011*). The last two are coupled to CapCone-2 toxins (*Figure 5—source data 2*). Notably, despite their structural diversity, these immunity proteins are all TM or lipoproteins and, like TipC (*Whitney et al., 2017*; *Klein et al., 2018*), are predicted to operate at the membrane or in the periplasm (*Figure 5N-Q*, *Figure 5—source data 2*). This suggests that they intercept their cognate Lipocone toxin domain outside cells or in the membrane rather than within the cell.

## Lipocone toxins in predator-prey and other interspecific conflicts

In contrast to polymorphic toxins, which are typically deployed in intraspecific conflict between competing strains of the same species, other toxin systems are deployed against more distantly related target cells, such as prey and eukaryotic hosts (*Aravind et al., 2012*). While some of these closely parallel polymorphic toxins in their domain architecture, they are usually distinguished by the lack of an accompanying immunity protein. The simplest of these systems are secreted Min-Wnt proteins from bacteria and fungi. These present just a standalone Min-Wnt domain or one fused to

a novel domain with a half β-barrel wrapping around a helix (*Figure 5R*, *Figure 5—source data 2*). These are probably deployed as diffusible toxins that target rival organisms in the environment.

Another architectural theme is defined by Min-Wnt and prok-SAA Lipocones fused to an enigmatic, novel, short C-terminal domain, which is comprised of a long β-hairpin with a characteristic "break" in its central region, causing it to acquire an arch-like appearance (*Figure 5S*, *Figure 5—figure supplement 3*). Hence, we refer to this domain as the broken-hairpin. We found the broken-hairpin domain to be fused to a wide array of predicted toxin domains across the bacterial superkingdom. These include effector domains otherwise found in polymorphic toxin and allied systems that target peptidoglycan, carrier lipids and the membrane, such as members of the Colicin M (*Chérier et al., 2021*), Zeta toxin-kinase (*Mutschler et al., 2011*), lysozyme (*Monzingo et al., 1996*), an α/β-hydrolase superfamilies (*Suplatov et al., 2012*) and nuclease toxins such as members of the HNH, HipA, SNase, and BECR superfamilies (*Zhang et al., 2012*; *Figure 5S*, *Figure 5—figure supplement 3C*). Remarkably, these proteins with the broken-hairpin tend to lack a signal peptide or association with any other secretion system or immunity proteins (*Figure 5S*, *Figure 5—source data 2*). Hence, we propose that the broken-hairpin domain itself serves as a trafficking mechanism for the externalization of these toxins in conflicts with rival environmental organisms.

Some predicted secreted Lipocones are found predominantly in predatory bacteria. The first of these are CapCone-2 domains from lineages like Bdellovibrionota, which are encoded in two-gene systems, with the second gene coding for a further secreted effector such as an α/β-hydrolase, Patatin, or acyltransferase or an OMP β-barrel domain (*Suplatov et al., 2012*; *Wimley, 2003*; *Fairman et al., 2011*; *Ghosh et al., 2006*; *Figure 5T*, *Figure 6—figure supplement 3*). Myxobacteria and some other lineages code for secreted prok-SAA domains fused to a N-terminal Zincin-like metallopeptidase domain, and the first bacterial example of the von Willebrand Factor D (vWD) and Ig domains at the C-terminus (*Dong et al., 2019*; *Figure 5T*, *Figure 5—source data 2*). In the recently described predatory Patescibacterial branch of Omnitrophota species, Skillet-clade Lipocone domains are found in gigantic proteins combined with several other domains and TM segments. Domains found in these proteins include polysaccharide biosynthesis enzymes (*Suresh Kumar et al., 2007*; *Rai and Mitchell, 2020*), signaling proteins involved in histidine kinase-receiver relays (*West and Stock, 2001*), peptidases of the MPTase and papain-like superfamily (*Novinec and Lenarčič, 2013*; *Dhanaraj et al., 1996*), diverse methylases, and extracellular ligand-binding domains like the peptidoglycan-binding LysM domain (*Costa et al., 2006*; *Figures 5T and 6*). Given the concentration of the above systems in predatory bacteria (*Figure 5—source data 2*), we posit that the above Lipocones might function as toxins targeting prey membranes alongside a battery of effectors targeting other cellular components. In particular, the CapCone-2 systems might play a role in the breaching of outer membranes by Bdellovibrionota. Animal vWD domains are involved in adhesion (*Zhang et al., 2018*); hence, the bacterial versions might play a similar role in adhering to prey cells, while the MPTase in these proteins potentially releases the associated Prok-SAA toxin through autoproteolysis. Finally, the giant proteins from the Patescibacteria are likely to combine signaling prey presence with overcoming prey defenses and breaching prey membranes.

Certain prok-TelC proteins are observed as part of several distinctive systems that could be involved in as-yet-undiscovered predatory interactions or in targeting environmental competitors. One such, defined by large proteins from spore-forming Bacillota, combines a diversifying set of extracellular ligand-binding domains (e.g., Ig-like, Cell-wall-binding β-hairpins and β-propellers *Williams and Barclay, 1988*; *Chen et al., 2011*; *Hermoso et al., 2003*) with a two-enzyme core formed by a prok-TelC and a N-acetylglucosamine (GlcNAc)–1-phosphodiester alpha-N-acetylglucosaminidase (NAGPA). NAGPA catalyzes phosphoric-diester hydrolysis to release phosphodiester-linked sugars (*Figures 5U and 6*, *Figure 6—figure supplement 3*; *Das et al., 2013*). Some of these proteins feature an additional NlpC/p60 superfamily peptidase domain predicted to target peptidoglycan (*Anantharaman and Aravind, 2003*). The recombinational diversity of ligand-binding domains in this system, even among closely related Bacillota species, supports a possible arms race and involvement in a biological conflict. Other TelC domains in some Bacillota, Actinomycetota, and fungi are fused to peptidoglycan-binding domains (PGBD) (*Dideberg et al., 1982*) and an Rv2525c-like TIM-barrel (*Bellinzoni et al., 2014*; *Figure 5U*, *Figure 6—figure supplement 3*). In Actinomycetota, this protein is further combined in operons with either of two mutually exclusive genes coding for rapidly evolving proteins (*Figure 5U*): (i) a secreted protein containing a pair of Ig domains (*Hermoso et al., 2003*)

(ii) a 3-TM protein (3TM-CCDN) with two conserved cysteines, an aspartate and asparagine residues predicted to be located between the TM segments outside the cell. This version is further coupled to a gene for a secreted VanY superfamily peptidase (*Arthur et al., 1992*; *Figure 5U*, *Figure 5—source data 2*). Common to these contexts are rapidly evolving and variable domains on the one hand and peptidoglycan/exopolysaccharide binding or degrading domains on the other. Hence, we interpret these as potential conflict systems that engage the cell wall and target it and associated membranes in rival bacteria.

## Lipocone domains in resistance to antimicrobial agents

VanZ-1 proteins (*Figure 1A*) were initially identified as encoded by a gene linked to that coding for the VanY D-alanyl-D-alanine carboxypeptidase involved in resistance to glycopeptide antibiotics like vancomycin and teicoplanin (*Arthur et al., 1995*; *Wright et al., 1992*; *Arthur et al., 1998*; *Arthur et al., 1994*; *Figure 6*). These antibiotics bind the terminal D-Ala-D-Ala in the peptide moiety of peptidoglycan, preventing the transpeptidase cross-linking reaction necessary for its maturation. Upon detection of these antibiotics, enzymes encoded by the core vancomycin resistance operon re-engineer the exported peptidoglycan by inserting a D-Ala-D-Lac in place of the D-Ala-D-Ala linkage, precluding antibiotic binding (*Stogios and Savchenko, 2020*). The VanY peptidase, while not strictly required for antibiotic resistance, acts as an accessory to this system by cleaving any remaining D-Ala-D-Ala linkages generated via the canonical pathway (*Stogios and Savchenko, 2020*; *Arthur et al., 1998*). However, the role of VanZ in this system has so far remained unknown. While only a small fraction of the VanZ-1 genes are found in these antibiotic resistance contexts (*Figure 5—source data 2*), interestingly, other Lipocone genes, namely those of the VanZ-2 and the Skillet families, might also be linked to VanY in lieu of VanZ-1. Furthermore, VanY might be replaced by a structurally unrelated secreted D-Ala-D-Ala carboxypeptidase of the metallo-beta-lactamase fold (*Palomeque-Messia et al., 1991*) in operonic contexts with VanZ-1 (*Figure 5V*). Hence, given our above prediction regarding VanZ acting in peptidoglycan and/or exopolysaccharide metabolism, VanZ-1 and the Lipocones displacing it might indeed play an accessory role with VanY at the membrane (*Wright et al., 1992*; *Arthur et al., 1998*) in antibiotic resistance. We posit that, in these contexts, it likely acts on the head group of Lipid II to recycle canonical peptidoglycan intermediates for their accelerated or more thorough replacement with the resistant versions (*Figure 3G*).

We also identified a conserved five-gene operon featuring a YfiM-1 family Lipocone that might play a role in resistance to antibacterial agents (*Figure 5W*, *Figure 6—figure supplement 1*). Other than YfiM-1, this operon contains genes for: (i) a thioredoxin domain protein *Qi and Grishin, 2005*; (ii) A DTW clade RNA modifying enzyme of the SPOUT superfamily *Meyer et al., 2016*; *Burroughs and Aravind, 2014*; (iii) a protein with acyl-CoA ligase, GNAT superfamily N-acetyltransferase and ATP-grasp domains *Dong et al., 2007*; *Fraser et al., 2002*; *Iyer et al., 2009*; (iv) a PssA-like phosphatidylserine synthetase of the HKD superfamily (*Raetz and Kennedy, 1974*; *Figure 5W*, *Figure 5—source data 2*). Of these enzymes, the phosphatidylserine synthetase is predicted to act in its usual capacity to generate a lipid with a serine head group (*Raetz and Kennedy, 1974*). We propose that this would then function as a substrate for the YfiM-1 Lipocone domain, which might exchange the serine for another moiety via a reaction paralleling PTDSS1/2 (*Figure 3A*). This moiety could then be modified by aminoacylation, further acylation, and a redox modification by the third protein listed above, together with the thioredoxin. Indeed, such peptide modifications of lipid head groups by lysine, alanine, or arginine aminoacylation catalyzed by derived tRNA synthetases fused to GNATs have been shown to be a key resistance mechanism against breaching of the membrane by antibacterial peptides (*Fields and Roy, 2018*; *Hancock, 1997*). Hence, we predict the modifications catalyzed by this system might play a comparable role. The presence of a tRNA-modifying DTW domain suggests that in parallel to the tRNA synthetases, the GNAT in this system might use a tRNA-linked acyl group as a substrate, as seen in peptidoglycan biosynthesis (*Benson et al., 2002*; *Schneider et al., 2004*).

## Eukaryotic recruitments of the Lipocone superfamily

Lipocone domains have been transferred on several occasions from bacteria to eukaryotes (*Figure 4*, *Figure 1—figure supplement 1*). While there is predicted functional overlap with the above-described, predominantly bacterial versions, we discuss these separately as the inferred biological contexts of their deployment are often distinct from the above.

## Plant YfiM-1 and eukaryotic VanZ-2 proteins

A conserved YfiM-1 family protein typified by the *Arabidopsis* AT1G15900 was acquired from the bacteroidetes lineage of bacteria at the base of the plant lineage prior to the chlorophyte-streptophyte (including land plants) split and is predicted to be catalytically active (*Figure 1—figure supplement 2*, *Figure 5—source data 2*). In *Arabidopsis*, this gene is widely expressed across different tissue types, developmental stages, and other tested conditions (*Schmid et al., 2003*; *Berardini et al., 2015*). Given the above-predicted roles for bacterial YfiM-1 proteins, it is conceivable that the plant version plays a comparable role in the metabolism of a conserved plant-specific lipid. In a similar vein, a distinct clade of standalone VanZ-2 domains typified by the *Saccharomyces cerevisiae* YJR112W-A was acquired early in the fungal lineage. A similar transfer is also seen in the SAR clade of eukaryotes (*Figure 1—figure supplement 1*). Since these eukaryotes lack peptidoglycan and other bacterial-type isoprenoid lipid-borne exopolysaccharide intermediates, we suggest that this version was recruited for modifications of a fungus-specific lipid (e.g., highly oxygenated isoprenoid lipids) (*Savidov et al., 2018*).

## The Met-TelC proteins

The Met-TelC clade is comprised of versions of the TelC family with a reconfigured active site transferred from bacteria to Metazoa prior to the divergence of the cnidarians, and most members are predicted to be catalytically inactive (*Figure 2*). In cnidarians and arthropods, the Met-TelC domain is found in a secreted protein fused to C-terminal adhesion-related vWA (*Lee et al., 1995*) and Ig domains, followed by a TM helix (*Figure 5X*, *Figure 5—source data 2*). The chordate version, typified by human PGLYRP2 (*Zhang et al., 2005*), is also secreted and is fused to a C-terminal amidase targeting the N-acetylmuramoyl-L-alanine linkage (*Figure 5X*, *Figure 5—source data 2*). PGLYRP2 is a key innate immunity factor against bacterial pathogens that degrade sugar-peptide linkages in peptidoglycan via the Amidase domain (*Lee et al., 2012*; *Dziarski and Gupta, 2010*; *Dziarski and Gupta, 2006b*). As most Met-TelC proteins lack the active site residues but are modeled to retain the substrate-binding pocket, we propose that they participate in anti-bacterial immunity as a Pathogen-Associated Molecular Pattern (PAMP) receptor (*Jones and Dangl, 2006*). Specifically, they could recognize polyisoprenoid pyrophosphate-linkage-containing lipid intermediates of bacterial cell-surface molecules like peptidoglycan or exopolysaccharides.

## Eukaryotic Wnt proteins

Wnt family Lipocones were transferred on multiple occasions to eukaryotes. The best-known of these are Met-Wnt proteins, which were acquired from bacteria at the base of Metazoa after they had separated from their closest sister group, the choanoflagellates. These lost the ancestral active site residues and function as well-studied secreted signaling molecules and will not be detailed further in this work (for review, see *Richards and Degnan, 2009*; *Holzem et al., 2024*). Independent of the Met-Wnt proteins, catalytically active, secreted versions closely related to the bacterial Min-Wnt proteins were transferred to fungi and, within Metazoa, to the rotifers and the hemichordate acorn worm *Saccoglossus kowalevskii*, where they are lineage-specifically expanded (*Figure 1—figure supplement 1*, *Figure 5—source data 2*). These versions are primarily standalone versions of the Min-Wnt domain, lacking the large inserts typical of the Met-Wnt proteins (*Figure 1A*). We predict that these eukaryotic Min-Wnt proteins retain their ancestral toxin role and might participate in anti-bacterial immunity.

## Met-SAA proteins

Met-SAA proteins (*Figure 1A*) were acquired from bacteria prior to the divergence of the cnidarians from the rest of Metazoa. However, unlike the Met-Wnt and Met-TelC proteins, they often conserve the ancestral active site residues, indicating that they are usually enzymatically active (*Figure 3*). Human SAA has been recognized as a key immune marker that dramatically increases in blood during the Acute Phase Response (*Morrow et al., 1981*). It has been reported to bind the *E. coli* outer membrane protein OmpA (*Hari-Dass et al., 2005*) and claimed to function as an opsonin in innate immunity (*Shah et al., 2006*). Like Met-TelC, but in contrast to Met-Wnts, Met-SAAs appear to have been lost or pseudogenized in several animal lineages (*Sack et al., 1989*; *Uhlar et al., 1994*; *Sun and*

*Ye, 2016*; *Figure 3*). This is consistent with an arms race scenario in immunity and the development of pathogen resistance against the Met-SAAs, leading to loss. Keeping with an immune role for the Met-SAAs, we propose a catalytic function for the active versions in severing lipid head groups of outer-membrane lipids or of isoprenoid lipid carrier intermediates. Such action could also generate PAMPs that could explain the activation of neutrophil- and macrophage-based immunity by SAA (*Shah et al., 2006*). Pertinent to these observations, diverse OMP β-barrels have been linked to the translocation of polymorphic toxin domains across the outer membrane of target cells (*Aoki et al., 2008*; *Virtanen et al., 2019*; *Ruhe et al., 2017*). Given this and the origin of Met-SAA from bacterial polymorphic toxin-related systems (*Figure 4*), its interaction with OmpA might help it cross over into the periplasmic space and act on maturing peptidoglycan or teichoic acid intermediates.

SAA was first reported as a component of secondary amyloid deposits (*Levin et al., 1972*), and its capacity to form amyloid fibrils upon protease cleavage was theorized as a potential PAMP activating the immune response (*Sack, 2018*). Indeed, bacteria produce their own secreted amyloids, such as Curli and Fap, believed to contribute to biofilm formation (*Blanco et al., 2012*; *Rouse et al., 2018*), and might be PAMPs recognized by animal immune systems (*Tükel et al., 2009*). Furthermore, other animal amyloids, such as the β-amyloid, have been proposed to play a role as physical barriers in immunity against bacteria (*Prosswimmer et al., 2024*). Thus, amyloid formation by protease cleavage (including potentially by bacterial proteases) may represent a second line of defense mediated by Met-SAA proteins.

## Discussion

### Early evolution of the lipocone superfamily

No single well-defined Lipocone clade is universally conserved across the three superkingdoms of Life (*Figure 4*, *Figure 1—figure supplement 1*, *Figure 1—figure supplement 5*). However, the VanZ and Wok clades are both found across all major bacterial phyla (notwithstanding sporadic losses in certain lineages) and in some archaeal lineages (*Figure 1—figure supplement 1*). At the same time, the cpCone clade is found across most major archaeal lineages and is nearly universally conserved in the eukaryotes (absent in Ascomycota and some choanoflagellates) (*Figure 1—figure supplement 1*). Notably, the cpCone and Wok clades tend to group together in the profile-profile similarity network (*Figure 1B*). These observations suggest that at least a single version of the Lipocone superfamily was likely present in the Last Universal Common Ancestor (LUCA). The phyletic patterns suggest that the LUCA Lipocone gave rise to the VanZ/Wok precursor in the bacterial lineage on the one hand and the cpCONE clade via a circular permutation event in the archaeo-eukaryotic lineage on the other (*Figure 4*). Based on the features of these deep-branching clades, the LUCA version is inferred to feature a hydrophobic domain with a 4TM helix core, with the active site facing the outer leaf of the lipid bilayer (*Figure 1C*). Given that extant versions operate both on classic phospholipids and isoprenoid lipids, it is difficult to infer which of these might have been substrates for the LUCA version. It is not impossible that this early version had a generic specificity that became specialized in the descendant clades.

### Subsequent diversification of the Lipocone domain

The early diversification of the Lipocone domain appears to have had different drivers in the two prokaryotic superkingdoms. The presence of an extensive repertoire of exopolysaccharides in the cell wall (peptidoglycan, teichoic acids), cell surface (e.g., ECA), and outer membrane (e.g., lipopolysaccharide), synthesized via isoprenoid lipid-linked intermediates, like lipid-II, was the primary driver in the bacterial superkingdom (*Egan et al., 2020*). Here, this diversification yielded four monophyletic groups: the VanZs, Wok, YfiM, and Skillet (*Figure 4*). The deeper VanZ and Wok branches, which were likely recruited first for lipid-II-related functions, were probably the predecessors of the more restricted bacterial families with specialized functions. For instance, the emergence of the outer membrane in certain bacteria was potentially coupled with the origin of the YfiM-like clade (*Figure 4*). Similarly, our predictions suggest that within these clades, further diversification accompanied the acquisition of specialized functional roles in antibiotic resistance, secondary sensor roles in single and multicomponent signaling, and lipoprotein processing. The interoperability of Lipocone domains on lipid carriers

shared across different biosynthetic pathways (see above, *Figure 3*, *Figure 6—figure supplement 1*, *Figure 6—figure supplement 2*) appears to have been a key factor leading to this versatility.

In the ancestral archaeo-eukaryotic lineage, the absence of peptidoglycan and an apparently lower diversity of structures with exopolysaccharides was reflected in the lesser diversification of the Lipocone clades (*Figure 4*). There are open questions regarding the biochemical functions of the primary archaeo-eukaryotic Lipocone clade, the cpCONE. Although the eukaryotic cpCone PTDSS1/2 family has been shown to swap serine for ethanolamine or choline in lipid head groups (*Stone and Vance, 1999*; *Vance, 2018*), their archaeal counterparts remain uncharacterized. Archaea have their own lipid with a serine in the head group (archaeophosphatidyserine), but to date, its synthesis has been shown to depend on a patchwork of different CDP-alcohol phosphatidyltransferase enzymes (CaPs) in different archaeal species (*Koga and Morii, 2005*; *Koga and Morii, 2007*; *Daiyasu et al., 2005*). While the CaPs are also integral membrane enzymes with a 6TM helix core, catalyzing comparable reactions as the Lipocones on lipid head groups in archaea and eukaryotes (*Daiyasu et al., 2005*), they are evolutionarily unrelated. Nevertheless, we suggest that the archaeal cpCones, like their eukaryotic counterparts, could contribute to distinct, as yet uncharacterized, pathways for the generation of cell membrane phospholipids like archaeophosphatidylserine or those with other head groups.

## Emergence of diffusible versions of the Lipocone domain and their repeated recruitment in biological conflicts

One of the remarkable aspects of the Lipocone superfamily is the loss of ancestral hydrophobicity in several families (*Figure 1C*), transforming them from integral membrane proteins to diffusible domains. While unexpected, such a transition in integral membrane enzymes acting on lipid substrates is not unprecedented. The PAP2 superfamily of integral membrane enzymes (e.g., diacylglycerol diphosphate phosphatase) (*Stukey and Carman, 1997*; *Carman and Han, 2006*) also contains several soluble versions (*Neuwald, 1997*) that appear to have emerged from an integral membrane ancestor (AMB and LA, unpublished observations). Most of the soluble Lipocone domains retain their active site conservation (*Figure 2*) and, at least in one experimentally characterized case, catalyze a comparable reaction as the TM version (*Whitney et al., 2017*; *Figure 3B*). The weight of the evidence presented here, including the profile-profile similarity network (*Figure 1B*), phyletic patterns (*Figure 1—figure supplement 1*), functional contexts (*Figures 5–6*), and the broadly shared structural features (*Figure 1A*, *Figure 1—figure supplement 2*, *Figure 1—figure supplement 4*), suggests that the loss of hydrophobicity occurred on a single occasion in the Lipocone superfamily, followed by diversification of these diffusible versions.

Our analysis of the diffusible Lipocone families reveals repeated recruitment as toxins/effectors in anti-viral and polymorphic toxin and allied systems (*Zhang et al., 2012*), suggesting that their diversification was driven by the arms races arising from the biological conflicts where they are deployed. Recruitment of a representative of the VanZ-1 family as a polymorphic toxin on rare occasions (*Figures 5N and 6*) suggests a possible evolutionary pathway for their recruitment as toxins: the effector version of Lipocones attacking lipids in competing bacteria likely emerged from an ancestral version that catalyzed endogenous lipid-head-group modifications on the same lipids in metabolic pathways. Once versions with reduced hydrophobicity emerged, they could be deployed as diffusible effectors that were shared across extracellular and intracellular conflict systems, a trend previously recognized in many other effector domains (*Aravind et al., 2022*).

## Repeated acquisition of Lipocones of bacterial origin by eukaryotes

Unlike bacteria, eukaryotes as a whole do not possess a rich repertoire of Lipocone domains. The PTDSS1/2 family, vertically inherited from the archaeal progenitor, is the only version that can be inferred as being present in the Last Eukaryotic Common Ancestor (*Figure 4*). However, distinct Lipocone families of ultimately bacterial provenance were acquired early and fixed in certain eukaryotic lineages: (i) YfiM-1 in the plant lineage; (ii) the fungal VanZ-2 domains typified by the *Saccharomyces cerevisiae* YJR112W-A; (iii) Met-Wnt (discussed further below) (*Figure 1—figure supplement 1*). The early fixation of these versions in the eukaryotic lineages possessing them suggests that they were recruited for definitive 'housekeeping' or developmental roles in the respective lineages. Beyond these, the fungal and metazoan lineages show more sporadically distributed versions, which have all been acquired from bacterial secreted-toxin or antiviral systems: (i) Min-Wnt independently in fungi

and certain Metazoa; (ii) SAA; (iii) TelC; the latter two are absent in the basal-most metazoans, the sponges, but are present in Cnidaria, suggesting a relatively early acquisition (*Figure 4*). The weight of the evidence suggests that they have retained certain aspects of the ancestral bacterial effector function for anti-pathogen immunity in eukaryotes. This is consistent with both their episodic loss and lineage-specific expansion, the tendency to show rapid sequence divergence and, in the Met-TelC family, loss of catalytic activity (*Figure 2*, *Figure 1—figure supplement 1*, *Figure 5—source data 2*).

This independent acquisition of at least three distinct Lipocone families in metazoan immunity from polymorphic and allied effector systems of prokaryotes points to a persistent evolutionary trend. Notably, the Lipocone domains participating in animal immunity have been drawn from secreted effectors rather than the intracellular versions (bacterial intracellular Min-Wnts) predicted to participate in bacterial anti-selfish element immunity. More generally, this adds to a growing list of components drawn from secreted effector systems of prokaryotes in eukaryotic immune systems (*Zhang et al., 2012*; *Aravind et al., 2012*; *Aravind et al., 2024*). For example, this closely parallels another structurally unrelated effector domain, the Zn-dependent deaminase (e.g., metazoan AID/APOBEC deaminases) (*Krishnan et al., 2018*). Hence, these observations add further support to our hypothesis that the extensive expansion of effectors in diverse prokaryotic inter-organismal conflict systems served as a reservoir from which eukaryotic immune systems repeatedly acquired components (*Aravind et al., 2012*; *Aravind et al., 2024*). We propose that symbiotic associations between the early animals and bacteria resulted in potential interactions via secreted effectors of the latter that aided the former against antagonistic bacteria. This probably led to their eventual acquisition by animals and incorporation into their immune processes.

## Origin of Wnt as a signaling molecule

Earlier considerations on the evolution of Wnt signaling indicated that it emerged at the base of the metazoan lineage and incorporated a wide range of components of different origins (e.g., the HMG domain transcription factor TCF/LEF, the HEAT repeat protein β-catenin and the 7TM receptor Frizzled) (*Richards and Degnan, 2009*). However, the provenance of Met-Wnt itself had been mysterious and was seen as a possible example of a metazoan innovation (*Holzem et al., 2024*). While the Met-Wnt domains possess peculiar structural elaborations (*Janda et al., 2012*), its conserved core is a Lipocone domain (*Figure 1A*). We establish that the progenitor of Met-Wnt emerged as part of the radiation of Lipocone domains in bacteria as effectors deployed in both intracellular and inter-organismal conflict – the Min-Wnt proteins.

Whereas the Min-Wnt proteins are predicted to be secreted toxins, the Met-Wnts underwent an ancestral inactivation through loss of the catalytic residues (*Figure 2*). However, they retained their ancient involvement in cell-cell interactions as secreted agents. The Met-Wnt residues recognized as essential for the receptor (Frizzled) binding, including the absolutely conserved palmitoleoylated serine residue, are found in the aforementioned Metazoa-specific hairpins and loops (*Janda et al., 2012*; *Zhong et al., 2021*). However, despite their inactivation, the Met-Wnts retain the ancestral substrate-binding pocket (*Figure 1A*, *Figure 1—figure supplement 2*). This raises the possibility that they might be involved in as-yet unexplored interactions with ligands such as lipids.

Our tracing of the provenance of Wnt back to an effector in secreted bacterial toxin systems adds it to a growing list of components in metazoan signaling networks that have been acquired from such systems. For instance, this is also the case with components of the other key metazoan signaling pathway, Hedgehog (*Zhang et al., 2011*). Here, the Hedgehog protein itself contains an autoproteolytic HINT peptidase domain that was likely drawn from a structurally and functionally cognate domain observed in polymorphic toxin systems (*Zhang et al., 2012*; *Zhang et al., 2011*). Further, an intracellular component of the same signaling pathway, Suppressor of Fused (SuFu), was derived from a common immunity protein found in polymorphic toxin systems (*Zhang et al., 2011*). Similarly, the Teneurin/Odd Oz proteins mediating signaling in cell migration, neuronal pathfinding, and fasciculation in Metazoa descended from a polymorphic toxin protein with a C-terminal HNH endonuclease toxin tip (*Zhang et al., 2012*). In a similar vein, the immunity protein of certain CapCone toxins identified in this study might have given rise to the β-sandwich domain in the eukaryotic centriolar assembly factor SAS-6. These observations suggest that, in addition to immune system components, interactions with symbiotic bacteria also potentially furnished the progenitors of components of eukaryotic signaling and cytoskeletal networks that were central to

the emergence of Metazoa as a clade of multicellular eukaryotes (*Kaur et al., 2020*; *Kaur et al., 2021*).

## Conclusions

Using sensitive sequence and structure analysis, we unify a large, hitherto unrecognized superfamily of enzymatic domains, the Lipocone. By combining analysis of the active site and the structure of the Lipocone domain with contextual information from conserved gene-neighborhoods and domain architectures, we present evidence that members of this superfamily target phosphate linkages in head groups of both classical phospholipids and polyisoprenoid lipids. Specifically, they catalyze reactions such as head group exchange or severing of the head group-diphosphate linkage from the polyisoprenol. We present evidence that these activities have been recruited in a wide range of biochemical contexts, including cell membrane lipid modification, metabolism of peptidoglycan and exopolysaccharide lipid-carrier linked intermediates, lipoprotein modifications, bacterial outer membrane modification, sensing of membrane-associated signals, effector activity in antiviral and inter-organismal conflicts, and resistance to antimicrobials. Furthermore, catalytically inactive versions like Met-Wnt have been recruited for signaling roles in Metazoa. We predict the catalytic activity and potential biochemical pathways of numerous representatives for the first time, including some proteins that have remained enigmatic for over two decades, like VanZ.

We identify three notable trends in Lipocone evolution. First, although we reconstruct the ancestral member of the superfamily as being a 4TM integral membrane domain, a large monophyletic subset underwent a dramatic loss of hydrophobicity, transforming them into diffusible versions, including the Wnts and the SAAs (*Figure 1C*). Second, the superfamily expanded in two major functional niches in bacteria, namely peptidoglycan/exopolysaccharide metabolism and effector domains of both secreted toxins and immune systems (*Figure 4*). Finally, members of the Lipocone superfamily were acquired on multiple occasions from bacteria by Metazoan and were reused in new functional contexts as signaling messengers and immune factors (*Figure 4*).

Importantly, our predictions in this regard underscore that much remains unexplored in terms of lineage-specific cell wall and membrane metabolism in prokaryotes. We present several testable biochemical, functional hypotheses for the many poorly understood branches of the superfamily, several of which are being recognized as enzymatic for the first time here. We hope this will also open new avenues of research to fill key gaps in our understanding of lipid metabolism.

## Methods
### Sequence analysis

Sequence similarity searches were performed using PSI-BLAST (*Altschul et al., 1997*) and JackHMMER (*Johnson et al., 2010*) against the NCBI non-redundant protein database (nr) (*Sayers et al., 2022*) or a version clustered down to 50% sequence identity (nr50). The searches were initiated using the previously identified prokaryotic Wnt (*Burroughs and Aravind, 2020*), with multiple rounds of searches conducted, each using seeds collected from the preceding searches (*Figure 1—figure supplement 5* file 5). Clusters based on sequence similarity (percentage identity or bit-score) were generated using MMseqs (*Hauser et al., 2016*). The clustering parameters were adjusted according to specific goals, enabling redundancy removal, the definition of homologous groups, and the creation of new profiles. Multiple sequence alignments (MSA) were generated using the MAFFT program (*Katoh and Standley, 2013*) with the local-pair algorithm, combined with the parameters –maxiterate 3000, –op 1.5, and –ep 0.2, and were manually refined based on structural superpositions and profile-profile comparisons.

### Sequence similarity network analysis

The HHalign program (*Steinegger et al., 2019*) was used to perform profile-profile comparisons, with the resulting p-value and e-value scores serving as edges for constructing a superfamily relationship network. This was then analyzed using the Leiden community finding algorithm (*Traag et al., 2019*) to detect sub-networks. Network analysis and visualization were performed using the R igraph (*Csardi and Nepusz, 2006*) or Python networkX libraries (*Hagberg et al., 2008*).

## Comparative genomics, domain identification, and phylogenetic analysis

Genomic neighborhoods were obtained from genomes available in the NCBI Genome database (*Sayers et al., 2022*) using in-house scripts written in Perl and Python. Conservation analysis of these genomic neighborhoods was performed by clustering the protein products of neighboring genes. Domain identification was conducted using a collection of HMMs and PSSMs maintained by the Aravind lab, along with HMMs from the Pfam database (*Finn et al., 2016*), utilizing the RPSBLAST (*Schäffer et al., 1999*) and HMMSCAN (*Eddy, 2011*) programs (*Figure 1—figure supplement 1—l* file 5). To further refine detection, domain identification was extended through remote homology analysis using the HHpred (*Söding et al., 2005*) program, against profiles built from the Pfam (*Finn et al., 2016*) and PDB70 (*Berman et al., 2007*) databases. Phylogenetic analyses were performed using FastTree (*Price et al., 2010*) and iqTREE2 (*Minh et al., 2020*). Experimental functional data for characterized members of the superfamily were collected with the assistance of the ChatGPT language model (https://chat.openai.com). Structural comparisons, along with shared genomic associations, were used to further refine the interrelationships within and between the groups of the superfamily.

Families with broader presence across multiple major lineages ('phyla') and deeper conservation within each of those lineages were inferred to be more ancient. In contrast, those with a more limited phyletic spread and/or limited depth of occurrence within each major lineage were likely later derivations (*Figure 4*, *Figure 1—figure supplement 1*). We formalized this inference by calculating a phyletic metric for the Lipocone clades (*Figure 1—figure supplement 1*) comprised of both the phyletic spread and depth. The phyletic spread $S_i$ of the $i$th Lipocone clade was computed thus:

$$S_i = \frac{m_i}{M},$$

Where $m_i$ is the number of lineages with at least one representative of the Lipocone clade $S_i$, and $M$ is the total number of lineages examined. The phyletic depth $D_i$ of the $i$th Lipocone clade was computed as a weighted average of its occurrence within each lineage in the form of the mediant:

$$D_i = \frac{\sum_{j=1}^{M} n_j}{\sum_{j=1}^{M} N_j},$$

where $n_j$ is the number of species in lineage $j$ with a Lipocone domain of the $i$th Lipocone clade and $N_j$ is the total number of species sampled in lineage $j$. $S_i$ and $D_i$ are plotted as a bar graph with $S_i$ as its width and $D_i$ its height.

## Contextual network construction

Each domain architecture and conserved gene neighborhood was decomposed into its constituent domains. These domains were then labeled for their biochemical function and stored as a YAML file (*Figure 6—source data 2*). The contextual connections were then rendered as a graph with the domains as its nodes and the adjacency relationships as its edges. Cliques containing a given Lipocone domain were detected in this graph and merged to constitute their respective dense subgraphs. These subgraphs were then examined for the statistically significant prevalence of particular labeled functions using the Fisher exact test. Network analysis was performed using the functions of the R igraph or Python networkX libraries.

## Structure analysis

Protein structures were modeled using Alphafold3 (*Abramson et al., 2024*), with visualization and manipulation performed using either MOL* (*Sehnal et al., 2021*) or PyMOL. Structural similarity searches were conducted using the DALIlite (*Holm, 2019*) and FOLDSEEK (*van Kempen et al., 2024*) programs. DALIlite was also used to generate structural alignments.

## Hydrophobicity analysis

To create the membrane propensity plots, for each protein $P_i$ in a given family, we compute the average TM-propensity of its amino acids using the TM tendency scale (*Zhao and London, 2006*). This score $H_i$ for $P_i$ is calculated as:

$$H_i = \frac{1}{n} \sum_{j=1}^{n} h_j$$

where $h_j$ is the TM tendency of the $j$-th amino acid in the protein $P_i$, and $n$ is its total length in amino acids (*Figure 1—source data 1*). The Kruskal–Wallis nonparametric test was applied to assess whether TM propensity scores differed across the 30 groups. As the Kruskal–Wallis test indicated a significant difference ($p<0.05$), we performed post-hoc pairwise comparisons using Dunn's test with Bonferroni correction to control for multiple testing. Group-wise visualizations were presented using critical difference diagrams, where groups not connected by horizontal bars are significantly different (adjusted $p<0.05$) (*Figure 1—figure supplement 3*).

## Acknowledgements

This research was supported by the Division of Intramural Research at the National Library of Medicine (NLM), National Institutes of Health (NIH). This research was supported in part by an appointment to the NLM Research Participation Program administered by the Oak Ridge Institute for Science and Education (ORISE) through an interagency agreement between the U.S. Department of Energy (DOE) and the NLM.

## Additional information

### Funding

| Funder | Grant reference number | Author |
| --- | --- | --- |
| Division of Intramural Research at the National Library of Medicine (NLM) | | A Maxwell Burroughs Gianlucca G Nicastro L Aravind |
| National Institutes of Health | | A Maxwell Burroughs Gianlucca G Nicastro L Aravind |
| Oak Ridge Institute for Science and Education | | Gianlucca G Nicastro |

The funders had no role in study design, data collection and interpretation, or the decision to submit the work for publication.

### Author contributions

A Maxwell Burroughs, Conceptualization, Data curation, Formal analysis, Validation, Investigation, Visualization, Methodology, Writing – original draft, Writing – review and editing; Gianlucca G Nicastro, Conceptualization, Data curation, Software, Formal analysis, Validation, Investigation, Visualization, Methodology; L Aravind, Conceptualization, Formal analysis, Supervision, Funding acquisition, Validation, Investigation, Methodology, Project administration, Writing – review and editing

### Author ORCIDs

A Maxwell Burroughs ⓘ http://orcid.org/0000-0002-2229-8771
Gianlucca G Nicastro ⓘ http://orcid.org/0000-0002-3133-2441
L Aravind ⓘ https://orcid.org/0000-0003-0771-253X

Reviewer #1 (Public review): https://doi.org/10.7554/eLife.108061.2.sa1
Reviewer #2 (Public review): https://doi.org/10.7554/eLife.108061.2.sa2
Reviewer #3 (Public review): https://doi.org/10.7554/eLife.108061.2.sa3
Author response https://doi.org/10.7554/eLife.108061.2.sa4

# Additional files

**Supplementary files**
MDAR checklist

**Data availability**
*Figure 1—source data 1*, *Figure 5—source data 1 and 2*, *Figure 6—source data 1* and *Figure 6—source data 2* contain the data used to generate the figures.

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
