## [Editor Report · eLife Assessment]

This **fundamental** study presents a **compelling** and comprehensive analysis of the newly defined Lipocone superfamily, offering unprecedented insights into the evolutionary origins of Wnt proteins. The authors provide evidence that this superfamily evolved from membrane proteins. The work is exemplary in its use of sequence analysis and structural modeling and will be of broad interest to researchers studying protein evolution and enzymology.

[Editors' note: this paper was reviewed by Review Commons.]

---

## [Referee Report · Reviewer #1 (Public review)]

Summary:

In this study titled 'The Lipocone Superfamily: A Unifying Theme In Metabolism Of Lipids, Peptidoglycan, And Exopolysaccharides, Inter-Organismal Conflicts And Immunity' from L. Aravind's group, the authors report the identification of a novel domain superfamily termed "Lipocone" superfamily. This superfamily unifies Wnt protein with a spectrum of domains from about 30 families, including those from phosphatidylserine synthases (PTDSS1/2), TelC toxin, VanZ proteins, and the animal Serum Amyloid A (SAA). The authors provide evidence that this superfamily originated as membrane proteins, with few (including Wnt and SAA) evolving into soluble domains. The authors also provide contextual evidence for the Lipocone members recruited as effectors in biological conflicts in both prokaryotes and eukaryotes. Importantly, to my knowledge, this study is the first to decipher the origins of Wnt signaling (emerging from a membrane protein context) and provide novel insights into immunity.

- The study is well-executed and provides many interesting leads for further experimental studies, which makes it very important. One of the significant hypotheses in this context is metazoan Wnt Lipocone domain interactions with lipids, which remain to be explored.

- The manuscript is generally navigable for interesting reading despite being content-rich.

- Overall, the figures are easy to follow.

Significance:

This study not only provides a plausible solution to the origins of metazoan Wnt signaling but also hypothesizes, based on retained ancestral substrate binding pocket, potential lipid interactions for lipocone wnt domains. The study also predicts novel enzymatic roles for many poorly characterized proteins that are involved in immunity across lineages/superkingdoms. This work is likely to inspire numerous experimental studies attempting to verify the hypotheses described in the study.

---

## [Referee Report · Reviewer #2 (Public review)]

Summary:

This is a remarkable study, one of a kind. The authors trace the entire huge superfamily containing Wnt proteins which origins remained obscure before this work. Even more amazingly, they show that Wnts originated from transmembrane enzymes. The work is masterfully executed and presented. The conclusions are strongly supported by multiple lines of evidence. Illustrations are beautifully crafted. This is an exemplary work of how modern sequence and structure analysis methods should be used to gain unprecedented insights into protein evolution and origins.

Significance:

Wnts are essential in animal development and their studies attracted significant attention. Therefore, this work is of high importance. Moreover, the authors delineated the entire superfamily consisting of many families with unique functional roles throughout all domains of life. The broad reach of this work further elevates its significance.

---

## [Referee Report · Reviewer #3 (Public review)]

Summary:

The manuscript by Burroughs et al. uses informatic sequence analysis and structural modeling to define a very large, new superfamily which they dub the Lipocone superfamily, based on its function on lipid components and cone-shaped structure. The family includes known enzymatic domains as well as previously uncharacterized proteins (30 families in total). Support for the superfamily designation includes conserved residues located on the homologous helical structures within the fold. The findings include analyses that shed light on important evolutionary relationships including a model in which the superfamily originated as membrane proteins where one branch evolved into a soluble version. Their mechanistic proposals suggest possible functions for enzymes currently unassigned. There is also support for the evolutionary connection of this family with the human immune system. The work will be of interest to those in the broad areas of bioinformatics, enzyme mechanisms, and evolution. The work is technically well performed and presented.

---

## [Author Response]

**Point-by-point description of the revisions**

**Reviewer #1 (Evidence, reproducibility and clarity):**
The study is well-executed and provides many interesting leads for further experimental studies, which makes it very important. One of the significant hypotheses in this context is metazoan Wnt Lipocone domain interactions with lipids, which remain to be explored.The manuscript is generally navigable for interesting reading despite being content-rich. Overall, the figures are easy to follow.

We thank the reviewer for the thoughtful and favorable assessment.

Major comments:I urge the authors to consider creating a first figure summarizing the broad approach and process involved in discovering the lipocone superfamily. This would help the average reader easily follow the manuscript.It will be helpful to have the final model/synthesis figure, which provides a take-home message that combines the main deductions from Fig 1c, Fig 4, Fig 5, and Fig 6 to provide an eagle's eye view (also translating the arguments on Page 38 last para into this potential figure).

We have generated a two-part figure that synthesizes these two requests, also in line with the recommendations made by Reviewer 3. Depending on the accepting Review Commons journal, we plan to either submit this as a graphical abstract/TOC figure (as suggested by Reviewer 3) or as a single figure. We prefer starting with the first approach as it will keep our figure count the same.

Minor comments:Fig 1C: The authors should provide a statistical estimate of the difference in transmembrane tendency scores between the "membrane" and "globular" versions of the Lipocone domains.

To address this, we calculated group-wise differences using the Kruskal-Wallis nonparametric test, followed by Dunn’s test with Bonferroni correction for a more stringent evaluation. The results of which are presented as a critical difference diagram in the new Supplementary Figure S3. The analysis is explained in the Methods section of the revised manuscript, and the statistically significant difference is mentioned in the text. This analysis identifies three groups of significantly different Lipocone families based on their transmembrane tendency: those predicted (or known) to associate with the prokaryotic membranes, those predicted to be diffusible, and a small number of families residing in eukaryotic ER membranes or bacterial outer membranes.

**Reviewer #2 (Evidence, reproducibility and clarity):**
This is a remarkable study, one of a kind. The authors trace the entire huge superfamily containing Wnt proteins which origins remained obscure before this work. Even more amazingly, they show that Wnts originated from transmembrane enzymes. The work is masterfully executed and presented. The conclusions are strongly supported by multiple lines of evidence. Illustrations are beautifully crafted. This is an exemplary work of how modern sequence and structure analysis methods should be used to gain unprecedented insights into protein evolution and origins.

We thank the reviewer for the positive evaluation of our work.

Minor comments.(1) In fig 1, VanZ structure looks rather different from the rest and is a more tightly packed helical bundle. It might be useful for the readers to learn more about the arguments why authors consider this family to be homologous with the rest, and what caused these structural changes in packing of the helices.

First, the geometry of an α-helix can be approximated as a cylinder, resulting in contact points that are relatively small. Fewer contact constraints can lead to structural variation in the angular orientations between the helices of an all α-helical domain, resulting in some dispersion in space of the helical axes. As a result, some of the views can be a bit confounding when presented as static 2D images. Second, of the two VanZ clades the characteristic structure similar to the other superfamily members is more easily seen in the VanZ-2 clade (as illustrated in supplementary Figure S2).

Importantly, the membership of the VanZ domains was recovered via significant hits in our sequence analysis of the superfamily. When the sequence alignments of the active site are compared (Figure 2), VanZ retains the conserved active site residue positions, which are predicted to reside spatially in the same location and project into an equivalent active site pocket as seen in the other families in the superfamily. Further, this sequence relationship is captured by the edges in the network in Figure 1B: multiple members of the superfamily show edges indicating significant relationships with the two VanZ families (e.g., HHSearch hits of probability greater than 90%; p<0.0001 are observed between VanZ-1 and Skillet-DUF2809, Skillet-1, Skillet-4, YfiM-1, YfiM-DUF2279, Wok, pPTDSS, and cpCone-1). Thus, they occupy relatively central locations in the sequence similarity network, indicating a consistent sequence similarity connection to multiple other families.

(2) Fig. 4 color bars before names show a functional role. How does the blue bar "described for the first time" fits into this logic? Maybe some other way to mark this (an asterisk?) could be better to resolve this sematic inconsistency.

We have shifted the blue bars into asterisks, which follow family names, now stated in the updated legend.

**Reviewer #3 (Evidence, reproducibility and clarity):**
The manuscript by Burroughs et al. uses informatic sequence analysis and structural modeling to define a very large, new superfamily which they dub the Lipocone superfamily, based on its function on lipid components and cone-shaped structure. The family includes known enzymatic domains as well as previously uncharacterized proteins (30 families in total). Support for the superfamily designation includes conserved residues located on the homologous helical structures within the fold. The findings include analyses that shed light on important evolutionary relationships including a model in which the superfamily originated as membrane proteins where one branch evolved into a soluble version. Their mechanistic proposals suggest possible functions for enzymes currently unassigned. There is also support for the evolutionary connection of this family with the human immune system. The work will be of interest to those in the broad areas of bioinformatics, enzyme mechanisms, and evolution. The work is technically well performed and presented.

We appreciate the positive evaluation of our work by the reviewer.

**Referees cross-commenting**
All the comments seem useful to me. I like Reviewer 1's suggestion for a flowchart showing the methodology. I think the summarizing figure suggested could be a TOC abstract, which many journals request.

To accommodate this comment (along with Reviewer 1’s comments), we have generated a two-part figure containing the methodology flowchart and the summary of findings. Combining the two provides some before-and-after symmetry to a TOC figure, while also avoiding further inflation of the figure count, which would likely be an issue at one or more of the Review Commons journals.

The authors may wish to consider the following points (page numbers from PDF for review):(1) It would be useful in Fig 1A, either in main text or the supporting information, to also have an accompanying topology diagram- I like the coloring of the helices to show the homology but the connections between them are hard to follow

We acknowledge the reviewer’s concern as one shared by ourselves. We have placed such a topology diagram in Figure 1A, and now refer to it at multiple points in the manuscript text.

(2) Page: 6- In the paragraph marked as an example- please call out Fig1A when the family mentioned is described (I believe SAA is described as one example)

We have added these pointers in the text, where appropriate.

(3) Page: 7- The authors state "these 'hydrophobic families' often evince a deeper phyletic distribution pattern than the less-hydrophobic families (Figure S1), implying that the ancestral version of the superfamily was likely a TM domain" there should be more explanation or information here - I am not certain from looking at FigS1 what a deeper phyletic distribution pattern means. Perhaps explaining for a single example? I also see that this important point is discussed in the conclusions- it is useful to point to the conclusion here.

Our use of the ‘deeper’ in this context is meant to convey the concept that more widely conserved families/clades (both across and within lineages) suggest an earlier emergence. In the Lipocone superfamily, this phylogenetic reasoning supports an evolutionary scenario where the membrane-inserted versions generally emerged early, while the solubilized versions, which are found in relatively fewer lineages, emerged later.

To address this objectively, we have calculated a simple phyletic distribution metric that combines the phyletic spread of a Lipocone clade with its depth within individual lineages, which is then plotted as a bargraph (Supplemental Figure S1). Briefly, this takes the width of the bar as the phyletic spread across the number of distinct taxonomic lineages and its height as a weighted mean of occurrence within each lineage (depth). The latter helps dampen the effects of sampling bias. In the resulting graph, lineages with a lower height and width are likely to have been derived later than those with a greater height and width. A detailed description clarifying this has been added to the Methods section of the revised manuscript. The results support two statements that are made in the text: (1) that the Wok and VanZ clades are the most widely and deeply represented clades in the superfamily, and (2) that the predicted transmembrane versions tend to be more widely and deeply distributed. We have also added a statement in the results with a pointer to Figure S1 to clarify this point raised by the referee.

(4) For figure 3 I would suggest instead of coloring by atom type- to color the leaving group red and the group being added blue so the reader can see where the moieties start and end in substrates and products

We have retained the atom type coloring in the figure for ease of visualizing the atom types. However, to address the reviewer’s concern, we have added dashed colored circles to highlight attacking and leaving groups in the reactions. The legend has been updated accordingly.

(5) Page: 13- The authors state "While the second copy in these versions is catalytically inactive, the H1' from the second duplicate displaces the H1 from the first copy," So this results in a "sort of domain swap" correct? It may be more clear to label both copies in Figure S3 upper right so it is easier for the reader to follow.

We have added these labels to the updated Figure S4 (formerly S3).

(6) The authors state "In addition to the fusion to the OMP β-barrel, the YfiM-DUF2279 family (Figure 5H) shows operonic associations with a secreted MltG-like peptidoglycan lytic transglycosylase (127,128), a lipid anchored cytochrome c heme-binding domain (129), a phosphoglucomutase/phosphomannomutase enzyme (130), a GNAT acyltransferase (131), a diaminopimelate (DAP) epimerase (132), and a lysozyme like enzyme (133). In a distinct operon, YfiM-DUF2279 is combined with a GT-A glycosyltransferase domain (79), a further OMP β-barrel, and a secreted PDZ-like domain fused to a ClpP-like serine protease (134,135) (Figure 5H)." this combination of enzymes sounds like those in the pathways for oligosaccharide synthesis which is cytoplasmic but the flippase acts to bring the product to the periplasm. Please make sure it is clear that these enzymes may act at different faces of the membrane.

We have made that point explicit in the revised manuscript in the paragraph following the above-quoted statement.

(7) Page: 21- the authors should remove the unpublished observations on other RDD domain or explain or cite them

The analysis of the RDD domain is a part of a distinct study whose manuscript we are currently preparing, and explaining its many ramifications would be outside the scope of this manuscript. Moreover, placing even an account of it in this manuscript would break its flow and take the focus away from the Lipocone superfamily. Further, its inclusion of the RDD story would substantially increase the size of the manuscript. However, it is commonly fused to the Lipocone domain; hence, it would be remiss if we entirely remove a reference to it. Accordingly, we retain a brief account of the RDD-fused Lipocone domains in the revised manuscript that is just sufficient to make the relevant functional case.

(8) Page: 34- The authors state "For instance, the emergence of the outer membrane in certain bacteria was potentially coupled with the origin of the YfiM and Griddle clades (Figure 4)." I don't see origin point indicated in figure 4 emergence of outer membrane- this may be helpful to indicate in some way- also I am not certain what the dashed circles in Fig 4 are indicating- its not in the legend?

This annotation has been added to the revised Figure 4, and the point of recruitment is indicated with a “X” sign, along with a clarification in the legend regarding the dashed circles.

(9) In terms of the hydrophobicity analysis, it would be good to mark on the plot (Fig 1C) one or two examples of lipocone members with known structure that are transmembrane proteins as a positive control

We have added these markers (colored triangles and squares for these families to the plot).

Grammar, typosPage: 3- abstract severance is an odd word to use for hydrolysis or cleavage

We have changed to “cleavage”.

Page: 5- "While the structure of Wnt was described over a decade prior" should read "Although the structure of ..."Page 7 - "One family did not yield a consistent prediction for orientation"- please state which familyPage: 8 "While the ancestral pattern is noticeably degraded in the metazoan Wnt (Met-Wnt) family, it is strongly preserved in the prokaryotic Min-Wnt family." Should read "Although the ancestral..."throughout- please replace solved with experimentally determined to be clear and avoid jargonPlease replace "TelC severs the link" with "TelC cleaves the bond "

We have made the above changes.

Page: 19- the authors state "a lipobox-containing synaptojanin superfamily phosphoesterase (125) and a secreted R-P phosphatase (126) (see Figure 6, Supplementary Data)" I was uncertain if the authors meant Fig S6 or they meant see Fig 6 and something else in supplementary data. Please fix.

In this pointer, we intended to flag the relevant gene neighborhoods in both Figures 5H and 6, as well as highlight the additional examples contained in the Supplementary Data. We have updated the point.